# PERTURBATION ANALYSIS OF NEURAL COLLAPSE

## ABSTRACT

Training deep neural networks for classification often includes minimizing the training loss beyond the zero training error point. In this phase of training, a "neural collapse" behavior has been observed: the variability of features (outputs of the penultimate layer) of within-class samples decreases and the mean features of different classes approach a certain tight frame structure. Recent works analyze this behavior via idealized unconstrained features models where all the minimizers exhibit exact collapse. However, with practical networks and datasets, the features typically do not reach exact collapse, e.g., because deep layers cannot arbitrarily modify intermediate features that are far from being collapsed. In this paper, we propose a richer model that can capture this phenomenon by forcing the features to stay in the vicinity of a predefined features matrix (e.g., intermediate features). We explore the model in the small vicinity case via perturbation analysis and establish results that cannot be obtained by the previously studied models. For example, we prove reduction in the within-class variability of the optimized features compared to the predefined input features (via analyzing gradient flow on the "central-path" with minimal assumptions), analyze the minimizers in the near-collapse regime, and provide insights on the effect of regularization hyperparameters on the closeness to collapse. We support our theory with experiments in practical deep learning settings.

## 1 INTRODUCTION

Modern classification systems are typically based on deep neural networks (DNNs), whose parameters are optimized using a large amount of labeled training data. Their training scheme often includes minimizing the training loss beyond the zero training error point (Hoffer et al., 2017; Ma et al., 2018; Belkin et al., 2019). In this terminal phase of training, a "neural collapse" (NC) behavior has been empirically observed when using either cross-entropy (CE) loss (Papyan et al., 2020) or mean squared error (MSE) loss (Han et al., 2022).

The NC behavior includes several simultaneous phenomena that evolve as the number of epochs grows. The first phenomenon, dubbed NC1, is decrease in the variability of the features (outputs of the penultimate layer) of training samples from the same class. The second phenomenon, dubbed NC2, is increasing similarity of the structure of the inter-class features' means (after subtracting the global mean) to a simplex equiangular tight frame (ETF). The third phenomenon, dubbed NC3, is alignment of the last layer's weights with the inter-class features' means. A consequence of these phenomena is that the classifier's decision rule becomes similar to nearest class center in feature space.

Many recent works attempt to theoretically analyze the NC behavior (Mixon et al., 2020; Lu & Steinerberger, 2022; Wojtowytsch et al., 2021; Fang et al., 2021; Zhu et al., 2021; Graf et al., 2021; Ergen & Pilanci, 2021; Ji et al., 2021; Galanti et al., 2021; Tirer & Bruna, 2022; Zhou et al., 2022; Thrampoulidis et al., 2022; Yang et al., 2022; Kothapalli et al., 2022). The mathematical frameworks are almost always based on variants of the unconstrained features model (UFM), proposed by Mixon et al. (2020), which treats the (deepest) features of the training samples as free optimization variables (disconnected from data or intermediate/shallow features). Typically, in these "idealized" models all the minimizers exhibit "exact collapse" (i.e., their within-class variability is exactly 0 and an exact simplex ETF structure is demonstrated) provided that arbitrary (but nonzero) level of regularization is used.

However, the features of DNNs are not free optimization variables but outputs of predetermined architectures that get training samples as input and have parameters (shared by all the samples) that are hard to optimize. Thus, usually, the deepest features demonstrate reduced "NC distance metrics" (such as within-class variability) compared to features of intermediate layers but do not exhibit convergence to an exact collapse. Indeed, as can be seen in any NC paper that presents empirical results, the decrease in the NC metrics is typically finite and stops above zero at some epoch (the margin depends on the dataset complexity, architecture, hyperparameter tuning, etc.).

In this paper, this issue is taken into account by studying a model that can force the features to stay in the vicinity of a predefined features matrix. By considering the predefined features as intermediate features of a DNN, the proposed model allows us to analyze how deep features progress from, or relate to, shallower features. We explore the model in the small vicinity case via perturbation analysis and establish results that cannot be obtained by the previously studied UFMs. Specifically, we prove reduction in the within-class variability of the optimized features compared to the predefined input features. To obtain this result (for arbitrary input features), we prove monotonic decrease of within-class variability along gradient flow on the "central-path" of a UFM with minimal assumptions (i.e., we drop the assumptions and modifications of the flow that Han et al. (2022) did to facilitate their analysis). Next, we provide a closed-form approximation for the model's minimizer. Then, focusing on the case where the input features matrix is already near collapse (e.g., the penultimate features of a well-trained DNN), we present a fine-grained analysis of our closed-form approximation, which provides insights on the effect of regularization hyperparameters on the closeness to collapse. We support our theory with experiments in practical deep learning settings.

## 2 Background and Problem Setup

Consider a classification task with $K$ classes and $n$ training samples per class. Let us denote by $\mathbf{y}_k \in \mathbb{R}^K$ the one-hot vector with 1 in its $k$-th entry and by $\mathbf{x}_{k,i} \in \mathbb{R}^p$ the $i$-th training sample of the $k$-th class. DNN-based classifiers can be typically expressed as

$$\text{DNN}_{\boldsymbol{\Theta}}(\mathbf{x}) = \mathbf{W}\mathbf{h}_{\boldsymbol{\theta}}(\mathbf{x}) + \mathbf{b},$$

where $\mathbf{h}_{\boldsymbol{\theta}}(\cdot) : \mathbb{R}^p \to \mathbb{R}^d$ (with $d \geq K$) is the feature mapping that is composed of multiple layers (with learnable parameters $\boldsymbol{\theta}$), and $\mathbf{W} = [\mathbf{w}_1, \ldots, \mathbf{w}_K]^\top \in \mathbb{R}^{K \times d}$ ($\mathbf{w}_k^\top$ denotes the $k$th row of $\mathbf{W}$) and $\mathbf{b} \in \mathbb{R}^K$ are the weights and bias of the last classification layer. The network's parameters $\boldsymbol{\Theta} = \{\mathbf{W}, \mathbf{b}, \boldsymbol{\theta}\}$ are usually learned by empirical risk minimization

$$\min_{\boldsymbol{\Theta}} \ \frac{1}{Kn} \sum_{k=1}^K \sum_{i=1}^n \mathcal{L}\left(\mathbf{W}\mathbf{h}_{\boldsymbol{\theta}}(\mathbf{x}_{k,i}) + \mathbf{b}, \mathbf{y}_k\right) + \mathcal{R}\left(\boldsymbol{\Theta}\right),$$

where $\mathcal{L}(\cdot, \cdot)$ is a loss function (e.g., CE or MSE[1]) and $\mathcal{R}(\cdot)$ is a regularization term.

Following the work of Mixon et al. (2020), in order to mathematically show the emergence of minimizers with NC structure, most of the theoretical papers have followed the "unconstrained features model" (UFM) approach, where the features $\{\mathbf{h}_{\boldsymbol{\theta}}(\mathbf{x}_{k,i})\}$ are treated as free optimization variables $\{\mathbf{h}_{k,i}\}$. Namely, they study problems of the form

$$\min_{\mathbf{W}, \mathbf{b}, \{\mathbf{h}_{k,i}\}} \ \frac{1}{Kn} \sum_{k=1}^K \sum_{i=1}^n \mathcal{L}\left(\mathbf{W}\mathbf{h}_{k,i} + \mathbf{b}, \mathbf{y}_k\right) + \mathcal{R}\left(\mathbf{W}, \mathbf{b}, \{\mathbf{h}_{k,i}\}\right).$$

One such example is the work in (Tirer & Bruna, 2022), which considered a setting with regularized MSE loss (which shares similarity with models in the matrix factorization literature (Koren et al., 2009; Chi et al., 2019), except the assumptions that $d \geq K$ and on the specific structure of $\mathbf{Y}$):

$$\min_{\mathbf{W}, \mathbf{H}} \ \frac{1}{2Kn}\|\mathbf{W}\mathbf{H} - \mathbf{Y}\|_F^2 + \frac{\lambda_W}{2K}\|\mathbf{W}\|_F^2 + \frac{\lambda_H}{2Kn}\|\mathbf{H}\|_F^2, \tag{1}$$

where $\mathbf{H} = [\mathbf{h}_{1,1}, \ldots, \mathbf{h}_{1,n}, \mathbf{h}_{2,1}, \ldots, \mathbf{h}_{K,n}] \in \mathbb{R}^{d \times Kn}$ is the (organized) unconstrained features matrix, $\mathbf{Y} = \mathbf{I}_K \otimes \mathbf{1}_n^\top \in \mathbb{R}^{K \times Kn}$ (where $\otimes$ denotes the Kronecker product) is its associated one-hot vectors matrix, and $\lambda_W$ and $\lambda_H$ are positive regularization hyperparameters. It was shown that

---

[1]Hui & Belkin (2021) have shown that training DNN classifiers with MSE loss is a powerful strategy whose performance is similar to training with CE loss.

*all* the (global) minimizers of this bias-free UFM exhibit an orthogonal collapse, as stated in the following theorem.[2]

**Theorem 2.1** (Theorem 3.1 in (Tirer & Bruna, 2022)). *Let $d \geq K$ and define $c := \sqrt{\lambda_H \lambda_W}$. If $c \leq 1$, then any global minimizer $(\mathbf{W}^*, \mathbf{H}^*)$ of Eq. 1 satisfies*

$$\mathbf{h}_{k,1}^* = \ldots = \mathbf{h}_{k,n}^* =: \overline{\mathbf{h}}_k^*, \quad \forall k \in [K],$$

$$\|\overline{\mathbf{h}}_1^*\|_2^2 = \ldots = \|\overline{\mathbf{h}}_K^*\|_2^2 =: \rho = (1-c)\sqrt{\frac{\lambda_W}{\lambda_H}},$$

$$\left[\overline{\mathbf{h}}_1^*, \ldots, \overline{\mathbf{h}}_K^*\right]^\top \left[\overline{\mathbf{h}}_1^*, \ldots, \overline{\mathbf{h}}_K^*\right] = \rho \mathbf{I}_K,$$

$$\mathbf{w}_k^* = \sqrt{\lambda_H / \lambda_W} \, \overline{\mathbf{h}}_k^*, \quad \forall k \in [K].$$

*If $c > 1$, then Eq. 1 is minimized by $(\mathbf{W}^*, \mathbf{H}^*) = (\mathbf{0}, \mathbf{0})$.*

In short, the theorem states that any minimizer $(\mathbf{W}^*, \mathbf{H}^*)$ of Eq. 1 obeys that $\mathbf{H}^* = \overline{\mathbf{H}} \otimes \mathbf{1}_n^\top$ for some $\overline{\mathbf{H}} \in \mathbb{R}^{d \times K}$, and $\mathbf{W}^* \overline{\mathbf{H}} \propto \overline{\mathbf{H}}^\top \overline{\mathbf{H}} \propto \mathbf{W}^* \mathbf{W}^{*\top} \propto \mathbf{I}_K$. It is not hard to show that $\overline{\mathbf{H}}^\top \overline{\mathbf{H}} = \rho \mathbf{I}_K$ implies that

$$\left(\overline{\mathbf{H}} - \overline{\mathbf{h}}_G^* \mathbf{1}_K^\top\right)^\top \left(\overline{\mathbf{H}} - \overline{\mathbf{h}}_G^* \mathbf{1}_K^\top\right) = \rho \left(\mathbf{I}_K - \frac{1}{K} \mathbf{1}_K \mathbf{1}_K^\top\right),$$

where $\overline{\mathbf{h}}_G^* = \frac{1}{K} \sum_{k=1}^K \overline{\mathbf{h}}_k^* = \frac{1}{K} \overline{\mathbf{H}} \mathbf{1}_K$ is the global mean. Namely, the "mean-subtracted features" collapse to a simplex ETF. From the structure of the problem and the theorem, we see that there are infinitely many minimizers of Eq. 1. Indeed, as can be deduced from the proof of Theorem 2.1 in (Tirer & Bruna, 2022): Taking any (partial) orthonormal matrix $\mathbf{R} \in \mathbb{R}^{d \times K}$ (i.e., $\mathbf{R}^\top \mathbf{R} = \mathbf{I}_K$), one can construct a minimizer for Eq. 1 simply by $\mathbf{H}^* = \sqrt{\rho(\lambda_W, \lambda_H)} \mathbf{R} \otimes \mathbf{1}_n^\top$ and $\mathbf{W}^* = \sqrt{\lambda_H / \lambda_W} \sqrt{\rho(\lambda_W, \lambda_H)} \mathbf{R}^\top$.

The existing literature includes other different UFM settings where *all* the minimizers exhibit NC structures (e.g., see (Lu & Steinerberger, 2022; Wojtowytsch et al., 2021; Zhu et al., 2021; Fang et al., 2021; Thrampoulidis et al., 2022)). However, as discussed in Section 1, all the previously studied UFMs are idealized and their results deviate from the situation in practical DNN training, where the features do not exhibit exact collapse (e.g., since deep layers cannot arbitrarily modify intermediate features that are far from being collapsed) and the setting of the hyperparameters affects the distance from NC structure.

In this paper, we consider a different model with the goal of better analyzing the real-world "near collapse" situation where "exact NC" cannot be reached. Motivated by Eq. 1, we consider the following model

$$\min_{\mathbf{W}, \mathbf{H}} f(\mathbf{W}, \mathbf{H}; \mathbf{H}_0) = \frac{1}{2Kn} \|\mathbf{W}\mathbf{H} - \mathbf{Y}\|_F^2 + \frac{\lambda_W}{2K} \|\mathbf{W}\|_F^2 + \frac{\lambda_H}{2Kn} \|\mathbf{H}\|_F^2 + \frac{\beta}{2Kn} \|\mathbf{H} - \mathbf{H}_0\|_F^2, \tag{2}$$

where $\mathbf{H}_0 \in \mathbb{R}^{d \times Kn}$ is an input features matrix, which is fixed, and $\beta$ is a positive hyperparameter that controls the distance of $\mathbf{H}$ from $\mathbf{H}_0$.

Let us discuss the motivation for studying this model. As before, we interpret $\mathbf{W}$ and $\mathbf{H}$ as the final weights and deepest features of the DNN, respectively. Clearly, for $\mathbf{H}_0 = \mathbf{0}$ this model reduces to Eq. 1 (with $\|\mathbf{H}\|_F^2$ regularized by $\lambda_H + \beta$). Furthermore, when $\mathbf{H}_0$ is nonzero, but already a minimizer of Eq. 1 (and thus has zero within-class variability and an orthogonal frame structure), the following statement is straightforward.

**Corollary 2.2.** *Let $d \geq K$, $\lambda_H \lambda_W < 1$, and let $(\mathbf{W}^*, \mathbf{H}^*)$ be a minimizer of Eq. 1. Then, the minimizer of $f(\mathbf{W}, \mathbf{H}; \mathbf{H}_0 = \mathbf{H}^*)$ (in Eq. 2) is unique[3] and it is given by $(\mathbf{W}^*, \mathbf{H}^*)$.*

---

[2]Note that the results in (Tirer & Bruna, 2022) are stated for $\lambda_W \leftarrow \frac{\lambda_W}{K}$ and $\lambda_H \leftarrow \frac{\lambda_H}{Kn}$ (i.e., their hyperparameters absorb the factors $1/K$ and $1/Kn$ that are used here). Scaling the terms in the objective according to the number of samples, as done in Eq. 1, agrees with what is done in practice (e.g., averaging the squared errors over the minibatch samples rather than summing them). Our scaling also highlights the independence of the minimizers' properties on $K$ and $n$.

[3]Note that in both Eq. 1 and Eq. 2 the minimizer w.r.t. $\mathbf{W}$ is a closed-form function of $\mathbf{H}$: $\mathbf{W}^*(\mathbf{H}) = \mathbf{Y}\mathbf{H}^\top(\mathbf{H}\mathbf{H}^\top + n\lambda_W \mathbf{I}_d)^{-1}$. As such, a minimizer $\mathbf{H}^*$ of either objective uniquely implies the associated $\mathbf{W}^* = \mathbf{W}^*(\mathbf{H}^*)$.

That is, Eq. 2 allows us to pick one of the minimizers of Eq. 1 by $\mathbf{H}_0$ and transfer its orthogonal collapse properties, which are stated in Theorem 2.1, to the minimizer of Eq. 2.

However, the usefulness of Eq. 2 comes from exploring cases with nonzero/non-collapsed $\mathbf{H}_0$. Indeed, while $\mathbf{H}$ can be interpreted as the deepest features of a DNN, here we interpret $\mathbf{H}_0$ as the features that are obtained in a shallower layer. In this case, $1/\beta$ can be understood as the complexity of the subnetwork from $\mathbf{H}_0$ to $\mathbf{H}$. We are particularly interested in the the large $\beta$ regime, $\beta \gg 1$, where $\mathbf{H}_0$ expresses penultimate features (only one layer before $\mathbf{H}$) that significantly constrain $\mathbf{H}$. In Appendix F we review practical DNNs where the distance between the deepest and penultimate features may be small or is even inherently small. In this paper we focus on this large $\beta$ regime, and provide mathematical reasoning for the empirical NC behavior that are not captured by previously studied UFMs, such as proving that the optimized $\mathbf{H}$ has smaller within-class variability than $\mathbf{H}_0$, and analyzing how perturbations from collapse of $\mathbf{H}_0$ can be mitigated by the minimizer of Eq. 2.

## 3 DECREASE IN WITHIN-CLASS VARIABILITY

As discussed above, while the features matrix $\mathbf{H}$ represents the output of a DNN's penultimate layer, the input matrix $\mathbf{H}_0$ can be interpreted as the features of a preceding layer. Several works have presented empirical settings where the within-class variability of the features, measured by some "NC1 metric", decreases across depth (Papyan et al., 2020; Tirer & Bruna, 2022; Galanti, 2022). The goal of this section is to prove such a phenomenon for the model stated in Eq. 2. The theory that we provide shows also monotonic decrease of the within-class variability (till exact collapse) along gradient flow on the "central-path" of the UFM stated in Eq. 1.

Let us begin with several definitions that will be used in this section. For a given set of $n$ features for each of $K$ classes, $\{\mathbf{h}_{k,i}\}$, we define the per-class and global means as $\overline{\mathbf{h}}_k := \frac{1}{n} \sum_{i=1}^n \mathbf{h}_{k,i}$ and $\overline{\mathbf{h}}_G := \frac{1}{Kn} \sum_{k=1}^k \sum_{i=1}^n \mathbf{h}_{k,i}$, respectively, as well as the mean features matrix $\overline{\overline{\mathbf{H}}} := [\overline{\mathbf{h}}_1, \ldots, \overline{\mathbf{h}}_K]$. Next, we define the within-class and between-class $d \times d$ covariance matrices

$$\mathbf{\Sigma}_W(\mathbf{H}) := \frac{1}{Kn} \sum_{k=1}^K \sum_{i=1}^n (\mathbf{h}_{k,i} - \overline{\mathbf{h}}_k)(\mathbf{h}_{k,i} - \overline{\mathbf{h}}_k)^\top,$$

$$\mathbf{\Sigma}_B(\mathbf{H}) := \frac{1}{K} \sum_{k=1}^K (\overline{\mathbf{h}}_k - \overline{\mathbf{h}}_G)(\overline{\mathbf{h}}_k - \overline{\mathbf{h}}_G)^\top.$$

The within-class variability collapse (NC1) can be expressed as $\mathbf{\Sigma}_W(\mathbf{H}) \to \mathbf{0}$ while $\mathbf{\Sigma}_B(\mathbf{H}) \nrightarrow \mathbf{0}$, where the limit takes place with increasing the training epoch, and $\mathbf{\Sigma}_B(\mathbf{H}) > 0$ filters degenerate cases such as $\mathbf{H} = \mathbf{0}$. Several papers considered in their experiments the metric $\frac{1}{K}\text{Tr}\left(\mathbf{\Sigma}_W(\mathbf{H})\mathbf{\Sigma}_B^\dagger(\mathbf{H})\right)$, where $\mathbf{\Sigma}_B^\dagger$ denotes the pseudoinverse of $\mathbf{\Sigma}_B$ (Papyan et al., 2020; Han et al., 2022; Zhu et al., 2021). Yet, we believe that considering the metric

$$\widetilde{NC}_1(\mathbf{H}) := \text{Tr}\left(\mathbf{\Sigma}_W(\mathbf{H})\right) / \text{Tr}\left(\mathbf{\Sigma}_B(\mathbf{H})\right) \tag{3}$$

is more amenable for theoretical analysis while capturing the desired nondegenerate collapse behavior.[4] Indeed, the trace of a covariance matrix equals zero if and only if the covariance matrix is a zero matrix (this follows from $\text{Cov}^2(X, Y) \le \text{Var}(X)\text{Var}(Y)$).

Recall that the minimizer w.r.t. $\mathbf{W}$ in Eq. 2 (and Eq. 1) has a closed-form expression that is a function of $\mathbf{H}$, which is given by $\mathbf{W}^*(\mathbf{H}) = \mathbf{Y}\mathbf{H}^\top(\mathbf{H}\mathbf{H}^\top + n\lambda_W \mathbf{I}_d)^{-1}$. Thus, the optimization in Eq. 2 is equivalent to

$$\mathbf{H}_{1/\beta} := \underset{\mathbf{H}}{\text{argmin}} \ \mathcal{L}(\mathbf{H}) + \frac{\beta}{2Kn}\|\mathbf{H} - \mathbf{H}_0\|_F^2$$

---

[4]The metric $\frac{1}{K}\text{Tr}\left(\mathbf{\Sigma}_W \mathbf{\Sigma}_B^\dagger\right)$ was considered in (Han et al., 2022). Yet, to state a result on this metric the authors claim (in the proof of Cor. 2) that a nonzero eigenvalue of $\mathbf{\Sigma}_W^{-1/2}\overline{\overline{\mathbf{H}}}\overline{\overline{\mathbf{H}}}^\top \mathbf{\Sigma}_W^{-1/2}$ equals the reciprocal of the associated nonzero eigenvalue of $\mathbf{\Sigma}_W^{1/2}(\overline{\overline{\mathbf{H}}}\overline{\overline{\mathbf{H}}}^\top)^\dagger \mathbf{\Sigma}_W^{1/2}$. However, this is not correct in general (due to the inherent rank deficiency of $\overline{\overline{\mathbf{H}}}\overline{\overline{\mathbf{H}}}^\top$). For example, for $\mathbf{\Sigma}_W^{1/2} = \begin{bmatrix} 2 & 1 \\ 1 & 2 \end{bmatrix}$ and $\overline{\overline{\mathbf{H}}}\overline{\overline{\mathbf{H}}}^\top = \begin{bmatrix} 1 & 0 \\ 0 & 0 \end{bmatrix}$, we have that the single nonzero eigenvalue of the former is $5/9$ while the single nonzero eigenvalue of the latter is $5$.

where $\mathcal{L}(\mathbf{H}) := \frac{1}{2Kn}\|\mathbf{W}^*(\mathbf{H})\mathbf{H} - \mathbf{Y}\|_F^2 + \frac{\lambda_W}{2K}\|\mathbf{W}^*(\mathbf{H})\|_F^2 + \frac{\lambda_H}{2Kn}\|\mathbf{H}\|_F^2$.

For large $\beta$, the minimizer $\mathbf{H}_{1/\beta}$ can be viewed as a backward/implicit gradient descent update from $\mathbf{H}_0$ with respect to the loss $\mathcal{L}$. This follows from rewriting the first order optimality condition as

$$\frac{\mathbf{H}_{1/\beta} - \mathbf{H}_0}{1/\beta} = -Kn\nabla\mathcal{L}(\mathbf{H}_{1/\beta}).$$

Observing that for $\beta \to \infty$ we have $\mathbf{H}_{1/\beta} \to \mathbf{H}_0$ (formally shown in Appendix B), the above equation can be written as $\frac{d\mathbf{H}_t}{dt}\big|_{t=0} = -Kn\nabla\mathcal{L}(\mathbf{H}_0)$, where we think of $t$ as $\beta^{-1}$. This naturally gives rise to the gradient flow

$$\frac{d\mathbf{H}_t}{dt} = -Kn\nabla\mathcal{L}(\mathbf{H}_t), \tag{4}$$

associated with the UFM in Eq. 1. This means that results on this flow can be translated to results on the minimizer of Eq. 2 in the large $\beta$ regime. Indeed, in Theorem 3.1 below, we show that $\widetilde{NC}_1(\mathbf{H})$ monotonically decreases along this flow, which implies that $\widetilde{NC}_1(\mathbf{H}_{1/\beta}) < \widetilde{NC}_1(\mathbf{H}_0)$ for large enough $\beta$ (see the statement in Corollary 3.2 below).

Note that a flow for an objective that is equivalent to $\mathcal{L}(\mathbf{H})$ with $\lambda_W = 0$ and $\lambda_H = 0$ has been studied in (Han et al., 2022), who called it the "central path". The motivation for studying such an objective, where the optimization variable $\mathbf{W}$ is replaced by the optimal $\mathbf{W}^*(\mathbf{H})$, comes from the empirical observation in (Han et al., 2022) that the gap $\|\mathbf{W}^*(\mathbf{H})\mathbf{H} - \mathbf{Y}\|_F^2 - \|\mathbf{W}\mathbf{H} - \mathbf{Y}\|_F^2$ is rather small (compared to each term) during the optimization process of practical DNNs.

We now state our result for gradient flow on the "central path" (which is proved in Appendix A).

**Theorem 3.1.** *Assume that $\lambda_W > 0$, $\lambda_H \geq 0$, and that $\mathbf{H}_0$ is non-collapsed (i.e., $\mathbf{\Sigma}_W(\mathbf{H}_0) \neq \mathbf{0}$). Then, along the gradient flow, which is stated in Eq. 4, we have that*

- $\widetilde{NC}_1(\mathbf{H}_t)$ *strictly decreases along the flow untill it reaches zero.*

- $t \mapsto e^{2\lambda_H t}\mathrm{Tr}(\mathbf{\Sigma}_W(\mathbf{H}_t))$ *decreases along the flow. In particular, when $\lambda_H > 0$, $\mathrm{Tr}(\mathbf{\Sigma}_W(\mathbf{H}_t))$ decays exponentially.*

- $t \mapsto e^{2\lambda_H t}\mathrm{Tr}(\mathbf{\Sigma}_B(\mathbf{H}_t))$ *strictly increases along the flow.*

**Remark.** Note that our gradient flow analysis has minimal assumptions. Unlike (Han et al., 2022), our flow does not assume zero global mean ($\overline{\mathbf{h}}_G = \mathbf{0}$), $\lambda_W = \lambda_H = 0$ and invertibility of $\mathbf{\Sigma}_W$. And most importantly, it does not include any engineered renormalization and projection of the gradient, contrary to the previous work. Thus, it is more similar to practical gradient descent optimization of DNNs. Our unmodified flow and minimal assumptions require a different, and more general, analysis with quite involved computations.[5]

Not only does Theorem 3.1 state a monotonic decrease toward 0 in the NC1 metric, it also provides a separation between the behavior of $\mathrm{Tr}(\mathbf{\Sigma}_W)$ and $\mathrm{Tr}(\mathbf{\Sigma}_B)$ along the flow. A strict separation is observed for $\lambda_H = 0$: $\mathrm{Tr}(\mathbf{\Sigma}_W)$ decreases while $\mathrm{Tr}(\mathbf{\Sigma}_B)$ increases. As gradient flow is often used as a proxy for analyzing gradient descent with a small step-size (Elkabetz & Cohen, 2021), if we overlook the difference between optimizing the UFM in Eq. 1 jointly w.r.t. $\mathbf{W}$ and $\mathbf{H}$ and restricting the optimization to the "central path" $(\mathbf{W}^*(\mathbf{H}), \mathbf{H})$, then our theory also provides a mathematical reasoning for the experiments on gradient descent in (Tirer & Bruna, 2022) that show monotonic decrease in within-class variability.

Finally, with our interpretation of $t$ as $\beta^{-1}$, the following Corollary is a direct consequence of Theorem 3.1 and the continuity of $\nabla\mathcal{L}(\mathbf{H})$ (see Appendix B for a formal proof).

**Corollary 3.2.** *Assume that $\mathbf{H}_0$ is non-collapsed. Then, there exists some constant $C = C(\mathbf{H}_0) > 0$ such that for $\beta > C$ we have that $\widetilde{NC}_1(\mathbf{H}_{1/\beta}) < \widetilde{NC}_1(\mathbf{H}_0)$.*

---

[5]In more detail, all the assumptions in (Han et al., 2022) (including continually renormalization the gradient) lead to the fact that only the singular values (and not the singular vectors) of an "SNR matrix" $\mathbf{\Sigma}_W^{-1/2}(\mathbf{H})\overline{\mathbf{H}}$ vary along their flow. However, since we do not make their assumptions, we do not have such a matrix whose singular bases are fixed along the flow and we need to approach the problem in a more general way.

Recall that in the large $\beta$ regime we can interpret $\mathbf{H}$ as features of DNN that are deeper than $\mathbf{H}_0$ but such that the architecture between $\mathbf{H}_0$ and $\mathbf{H}$ is extremely simple (e.g., they are features of adjacent layers) and thus the distance between them is constrained. Under this interpretation, Corollary 3.2 implies that layer-wise optimization of DNN where each time a new layer is added (so that the previous deepest features $\mathbf{H}_{1/\beta}$ are considered as the new $\mathbf{H}_0$) will result in gradually depthwise decreasing NC1. An extension of the model in Eq. 2 that will include multiple levels of optimizable parameters may be able to provide similar reasoning to the gradual depthwise decrease in NC1 that is observed in practical DNN training, where all the layers are optimized simultaneously.

## 4 ANALYSIS OF THE NEAR-COLLAPSE REGIME

In this section, we will explore the behavior of the minimizers of Eq. 2 in the near collapse regime. As stated in Corollary 2.2, if $\mathbf{H}_0$ is already collapsed then the minimizer of Eq. 2 is also collapsed. This is aligned with the rationale that if we have a DNN that already exhibits collapse at some intermediate layer, we would expect the subsequent layers to maintain this collapse.[6] Essentially, we would like to analyze the minimizer of Eq. 2 for $\mathbf{H}_0$ that is not already collapsed. Unfortunately, for general non-collapsed $\mathbf{H}_0$ it is not likely that the minimizer is amenable for explicit analytical characterization. Yet, the fact that for orthogonally collapsed $\mathbf{H}_0 = \mathbf{H}^*$ we get a unique minimizer $(\mathbf{W}^*, \mathbf{H}^*)$ of Eq. 2, which is still characterized by Theorem 2.1, gives us a desirable setting for examining the minimizer of Eq. 2 obtained for $\tilde{\mathbf{H}}_0 = \mathbf{H}^* + \delta\mathbf{H}_0$ (with sufficiently small $\delta\mathbf{H}_0$) by exploiting our knowledge on $(\mathbf{W}^*, \mathbf{H}^*; \mathbf{H}_0 = \mathbf{H}^*)$. Analyzing the near-collapse setting will shed light on the way that the deviation from collapse in the input features is transferred to the optimized features, e.g., the amount of interaction within/between classes and the effects of hyperparameters. Such insights can be latter examined empirically beyond the near-collapse regime.

Let us denote by $(\tilde{\mathbf{W}}^*, \tilde{\mathbf{H}}^*)$ the minimizer of $f(\mathbf{W}, \mathbf{H}; \tilde{\mathbf{H}}_0)$. We are interested in studying the dependence of $\delta\mathbf{W} := \tilde{\mathbf{W}}^* - \mathbf{W}^*$ and $\delta\mathbf{H} := \tilde{\mathbf{H}}^* - \mathbf{H}^*$ on $\delta\mathbf{H}_0 = \tilde{\mathbf{H}}_0 - \mathbf{H}^*$ without the requirement of computing $(\tilde{\mathbf{W}}^*, \tilde{\mathbf{H}}^*)$ (that lack analytical expressions). In particular, our focus is on the relation between the features $\delta\mathbf{H}$ and $\delta\mathbf{H}_0$ (rather than $\delta\mathbf{W}$ and $\delta\mathbf{H}_0$), both because a minimizer $\tilde{\mathbf{H}}^*$ uniquely implies the associated $\tilde{\mathbf{W}}^*$, and because important aspects of NC, such as within-class variability decrease (NC1) and inter-class feature structure (NC2), consider the feature mapping rather than the last layer weights.

We begin with establishing such a result in the following theorem (which is proved in Appendix C) for $\mathbf{H}_0$ that is not necessarily a collapsed features matrix.

The notation in the theorem is as follows. We use $\text{vec}(\cdot)$ to denote the column-stack vectorization of a matrix. The derivatives are w.r.t. the vectorized matrices $\text{vec}(\mathbf{H})$ and $\text{vec}(\mathbf{W})$. For example, $\nabla_H f \in \mathbb{R}^{dnK \times 1}$ stands for the derivative of $f$ w.r.t. $\text{vec}(\mathbf{H})$, and a second derivative w.r.t. $\text{vec}(\mathbf{W})^\top$ yields $\nabla_W^\top \nabla_H f \in \mathbb{R}^{dnK \times Kd}$.

**Theorem 4.1.** *Let $d \geq K$, and set some $\mathbf{H}_0$ and $\delta\mathbf{H}_0$. Let $(\hat{\mathbf{W}}^*, \hat{\mathbf{H}}^*)$ be the minimizer of $f(\mathbf{W}, \mathbf{H}; \mathbf{H}_0)$ (with $f$ stated in Eq. 2). Let $(\tilde{\mathbf{W}}^*, \tilde{\mathbf{H}}^*)$ be the minimizer of $f(\mathbf{W}, \mathbf{H}; \tilde{\mathbf{H}}_0 = \mathbf{H}_0 + \delta\mathbf{H}_0)$. Define $\delta\mathbf{W} := \tilde{\mathbf{W}}^* - \hat{\mathbf{W}}^*$ and $\delta\mathbf{H} := \tilde{\mathbf{H}}^* - \hat{\mathbf{H}}^*$. Then, with approximation accuracy of $O(\|\delta\mathbf{H}\|^2, \|\delta\mathbf{W}\|^2, \|\delta\mathbf{H}_0\|^2)$, we have that*

$$\text{vec}(\delta\mathbf{H}) \approx \frac{\beta}{Kn} \left( \nabla_H^\top \nabla_H f - \nabla_W^\top \nabla_H f (\nabla_W^\top \nabla_W f)^{-1} \nabla_H^\top \nabla_W f \right)^{-1} \text{vec}(\delta\mathbf{H}_0),$$

$$\text{vec}(\delta\mathbf{W}) \approx -\frac{\beta}{Kn} (\nabla_W^\top \nabla_W f)^{-1} \nabla_H^\top \nabla_W f \left( \nabla_H^\top \nabla_H f - \nabla_W^\top \nabla_H f (\nabla_W^\top \nabla_W f)^{-1} \nabla_H^\top \nabla_W f \right)^{-1} \text{vec}(\delta\mathbf{H}_0),$$

*where all the derivatives[7] are evaluated at the point $(\hat{\mathbf{W}}^*, \hat{\mathbf{H}}^*; \mathbf{H}_0)$.*

*In particular, for $\beta \gg \max\{1, \lambda_H\}$ we have (with additional approximation error of $O(\beta^{-2})$)*

$$\text{vec}(\delta\mathbf{H}) \approx \left( \mathbf{I}_{dnK} - \frac{\lambda_H}{\beta} \mathbf{I}_{dnK} - \frac{1}{\beta} \mathbf{I}_{nK} \otimes \hat{\mathbf{W}}^{*\top} \hat{\mathbf{W}}^* + \frac{1}{\beta} \mathbf{Z}^* \right) \text{vec}(\delta\mathbf{H}_0), \tag{5}$$

---

[6] This is also aligned with empirical observations of gradual depthwise collapse in practical DNNs and with Corollary 3.2 at the limit where $\mathbf{H}_0$ is nearly collapsed.

[7] The derivatives are stated in the proof.

*where*

$$\mathbf{Z}^* := (\mathbf{E}^{*\top} + \hat{\mathbf{H}}^* \otimes \hat{\mathbf{W}}^*)^\top (\hat{\mathbf{H}}^* \hat{\mathbf{H}}^{*\top} \otimes \mathbf{I}_K + n\lambda_W \mathbf{I}_{dK})^{-1} (\mathbf{E}^{*\top} + \hat{\mathbf{H}}^* \otimes \hat{\mathbf{W}}^*),$$

$$\mathbf{E}^* := \big[ \mathrm{vec}(\mathbf{e}_{d,1}\mathbf{e}_{K,1}^\top(\hat{\mathbf{W}}^*\hat{\mathbf{H}}^* - \mathbf{Y})), ..., \mathrm{vec}(\mathbf{e}_{d,1}\mathbf{e}_{K,K}^\top(\hat{\mathbf{W}}^*\hat{\mathbf{H}}^* - \mathbf{Y})), \mathrm{vec}(\mathbf{e}_{d,2}\mathbf{e}_{K,1}^\top(\hat{\mathbf{W}}^*\hat{\mathbf{H}}^* - \mathbf{Y})), ...$$

$$..., \mathrm{vec}(\mathbf{e}_{d,d}\mathbf{e}_{K,K}^\top(\hat{\mathbf{W}}^*\hat{\mathbf{H}}^* - \mathbf{Y})) \big],$$

*and $\mathbf{e}_{d,i}$ is the standard vector in $\mathbb{R}^d$ with 1 in its $i$th entry (similar definition stands for $\mathbf{e}_{K,k}$).*

Observe that, assuming small approximation error, Theorem 4.1 states the linear operation that transforms $\delta\mathbf{H}_0$ to $\delta\mathbf{H}$. We will focus on the large $\beta$ regime that is stated in Eq. 5, where the matrix inversion can be well approximated. Furthermore, due to the vectorization operation, observe that the linear expression $\mathrm{vec}(\delta\mathbf{H}) \approx \mathbf{F}\mathrm{vec}(\delta\mathbf{H}_0)$ has the following block-based representation

$$\begin{bmatrix} \mathrm{vec}(\delta\mathbf{H}^{(1)}) \\ \vdots \\ \mathrm{vec}(\delta\mathbf{H}^{(K)}) \end{bmatrix} \approx \begin{bmatrix} \mathbf{F}_{1,1} & \dots & \mathbf{F}_{1,K} \\ & \ddots & \\ \mathbf{F}_{K,1} & \dots & \mathbf{F}_{K,K} \end{bmatrix} \begin{bmatrix} \mathrm{vec}(\delta\mathbf{H}_0^{(1)}) \\ \vdots \\ \mathrm{vec}(\delta\mathbf{H}_0^{(K)}) \end{bmatrix}, \tag{6}$$

where $\delta\mathbf{H}^{(k)} := \delta\mathbf{H}[:, dn(k-1)+1 : dnK] \in \mathbb{R}^{d \times n}$ is the sub-matrix of $\delta\mathbf{H}$ that is composed of the columns associated with the $k$th class (and similarly for $\delta\mathbf{H}_0$). Namely, we have that $\mathbf{F} \in \mathbb{R}^{dnK \times dnK}$ is composed of blocks of size $dn \times dn$. The diagonal blocks are the "intra-class blocks". Each of them shows the effect of perturbation in a certain class in $\mathbf{H}_0$ on the features of the same class in $\mathbf{H}$. The off-diagonal blocks are the "inter-class blocks". Each of them shows the effect of perturbation in a certain class in $\mathbf{H}_0$ on the features of another class in $\mathbf{H}$.

Recall that for $\mathbf{H}_0 = \mathbf{H}^*$ that is already exactly collapsed, the minimizer of $f(\cdot; \mathbf{H}_0)$ is also collapsed, so $\hat{\mathbf{H}}^* = \mathbf{H}^*$ in the above theorem. Importantly, in this case the matrix in Eq. 6 transforms deviation from exact collapse in the input features to deviation from exact collapse in the optimized features. Thus, we have that stronger attenuation behavior of the blocks of $\mathbf{F}$ (e.g., small singular values) implies that the minimizer $\tilde{\mathbf{H}}^*$ is closer to exact collapse. Based on specializing Theorem 4.1 to the near-collapse case, we present in the following theorem (which is proved in Appendix D) an exact analysis of singular values of the blocks of $\mathbf{F}$. (The notations $\sigma_{max}(\cdot)$ and $\sigma_{min}(\cdot)$ stand for the largest and smallest singular values of a matrix, respectively).

**Theorem 4.2.** *Consider the setting of Theorem 4.1, $\lambda_H\lambda_W < 1$ (assumed in Theorem 2.1), $d > K$, $\beta \gg \max\{1, \lambda_H\}$, and the representation of Eq. 5 that is given in Eq. 6. Let $\mathbf{H}_0$ be a collapse features matrix (minimizer of Eq. 1 for the same $\lambda_H, \lambda_W$ as in Eq. 2). Then, for $k, \tilde{k} \in [K]$ with $k \neq \tilde{k}$ we have that $\mathbf{F}_{k,k}$ is full rank, $\mathbf{F}_{k,\tilde{k}}$ is rank-1, and*

$$\sigma_{max}(\mathbf{F}_{k,k}) = 1,$$
$$\sigma_{min}(\mathbf{F}_{k,k}) = 1 - \beta^{-1}\sqrt{\lambda_H/\lambda_W},$$
$$\sigma_{max}(\mathbf{F}_{k,\tilde{k}}) = 2\beta^{-1}\lambda_H(1 - \sqrt{\lambda_H\lambda_W}).$$

**Remark.** In Appendix D we derive expressions for the complete singular value decomposition of $\mathbf{F}_{k,k}$ and $\mathbf{F}_{k,\tilde{k}}$. Our expressions for the entire spectrum of $\mathbf{F}_{k,k}$ reveal its step-wise decreasing shape, as visualized in Figure 1 for $\beta = 100, K = 4, d = 10, n = 10, \lambda_W = \sqrt{2}$ and various values of $\lambda_H$. To keep the paper concise, we state in the above theorem only the results for the maximal and minimal singular values of $\mathbf{F}_{k,k}$, but note that, similarly to $\sigma_{min}(\mathbf{F}_{k,k})$, almost all singular values decrease as $\lambda_H$ increases. Even though a small portion ($\frac{1-K/d}{n}$) of the singular values equal 1 (as shown in our analysis in Appendix D), we can still gain insights on the attenuation profile since generic perturbations are unlikely to concentrate in such an extremely low-dimensional subspace (and, in fact, the singular vectors associated with this subspace do not affect the within-class variability).

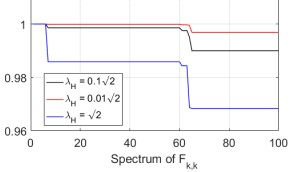

Figure 1: The effect of $\lambda_H$ on the spectrum of $\mathbf{F}_{k,k}$.

From Theorem 4.2 we gain the following insights on the minimizer of Eq. 2 in the near-collapse and large $\beta$ regime. First, observe that not only do exactly collapsed minimizers have orthogonal features

for different classes, but also in the near-collapse setting an intra-class block is much more dominant than each inter-class block, as follows from $\mathbf{F}_{k,\tilde{k}}$ being rank-1 and $\sigma_{max}(\mathbf{F}_{k,\tilde{k}}) \ll \sigma_{min}(\mathbf{F}_{k,k})$. For generic perturbations that do not concentrate in specific low-dimensional subspaces this implies that also before/near pure collapse, we have that the deviation from collapse in the features of a certain class is mainly due to deviation from collapse of input (preceding) features of the same class and not those of the $K-1$ other classes. (See Appendix D.1 for more details, and note that this also implies preservation of per-class near-collapse). Second, we see that the feature mapping regularization plays the major role in approaching (near-)collapse behavior. Indeed, increasing $\lambda_H$ decreases the spectral values of the (more dominant) intra-class blocks $\{\mathbf{F}_{k,k}\}$ (contrary to increasing $\lambda_W$). Recall that reducing the singular values of the blocks of $\mathbf{F}$ implies reducing the distance of the minimizer $\tilde{\mathbf{H}}^*$ from exact collapse. Third, our result on the inter-class blocks $\{\mathbf{F}_{k,\tilde{k}\neq k}\}$ hints that the regularization of the last layer's weights (determined by $\lambda_W > 0$) may still have a supportive effect on reaching (near-)collapse behavior by reducing the component of the deviation from collapse that is due to "crosstalk"/interference of features of different classes (e.g., when some classes are harder to be classified then others). In the sequel, we show that the above observations correlate with the NC behavior in practical settings.

## 5 EXPERIMENTS

In this section, we translate the insights that are obtained for the model in Eq. 2 to what is observed with practical DNNs and datasets. We evaluate the distance of DNN's features from exact NC using metrics that have been also used in previous works. Despite defining the metric $\widetilde{NC}_1$ in Eq. 3, here we mainly measure within-class variability using $NC_1 := \frac{1}{K}\mathrm{Tr}\left(\mathbf{\Sigma}_W \mathbf{\Sigma}_B^\dagger\right)$, where we use the definitions of Section 3. (We use this metric due to its popularity even though it is less amenable for theoretical analysis). We measure the structure of the features using

$$NC_2 := \left\| \frac{(\overline{\mathbf{H}} - \overline{\mathbf{h}}_G \mathbf{1}_K^\top)^\top (\overline{\mathbf{H}} - \overline{\mathbf{h}}_G \mathbf{1}_K^\top)}{\|(\overline{\mathbf{H}} - \overline{\mathbf{h}}_G \mathbf{1}_K^\top)^\top (\overline{\mathbf{H}} - \overline{\mathbf{h}}_G \mathbf{1}_K^\top)\|_F} - \frac{1}{\sqrt{K-1}}(\mathbf{I}_K - \frac{1}{K}\mathbf{1}_K \mathbf{1}_K^\top) \right\|_F,$$

where the simplex ETF is normalized to unit Frobenius norm.

The result of Section 3 provides reasoning to justify depthwise decrease in within-class variability, which has already been empirically demonstrated for end-to-end training in several papers (Papyan et al., 2020; Tirer & Bruna, 2022; Galanti, 2022) (we present such experiments in Appendix E.2). Here we show this behavior also for layer-wise training, which is better represented by our model. We consider the CIFAR-10 dataset and train an MLP with 1 to 10 hidden layers and a final classification layer. Each time, we add and train a hidden layer on top of the previous hidden layers, which are maintained fixed. Then we compute the NC1 metrics for the deepest features. Due to space limitation, the

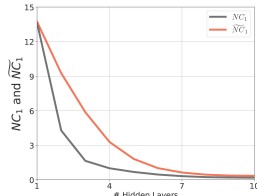

Figure 2: Layer-wise training of MLP on CIFAR-10.

experimental details are deferred to Appendix E.1. Figure 2 demonstrates decrease in both $NC_1$ and $\widetilde{NC}_1$ as we add more hidden layers on top the previous, which are maintained fixed. Note that our theory justifies such decrease for all the layers (the features are not required to be near collapse).

Next, we turn to demonstrate correlation of practical NC behavior with the insight gained in Section 4 that $\lambda_H$ plays a bigger role than $\lambda_W$ does in approaching NC. Based on the equivalence of $L_2$-regularization with weight decay (WD) in gradient-based methods, we can make the analogy of regularizing $\mathbf{H}$ in Eq. 2 to WD of the weights of practical DNNs in the feature mapping layers (i.e., excluding the last layer's weights). Importantly, note that this analogy is empirically justified for plain UFMs in (Zhu et al., 2021). Under this analogy, our analysis suggests that, as long as entering the zero training error phase of training is maintained, increasing (resp. decreasing) the WD in the feature mapping layers should decrease (resp. increase) the distance from exact collapse more than increasing (resp. decreasing) the WD in the classification layer. Indeed, we empirically show this behavior below. (More experiments are presented in Appendix E.2). We note that there exists a work that *empirically*[8] shows that WD facilitates collapse (Rangamani & Banburski-Fahey, 2022), however, they do not examine the WD in feature mapping and classification layers separately.

---

[8]Note that the claim in (Rangamani & Banburski-Fahey, 2022) that NC solution cannot minimize unregularized bias-free MSE loss comes from demanding that $\mathbf{H}^*$ – without subtracting the global mean – will be a simplex ETF rather than an orthogonal frame as shown in Theorem 2.1.

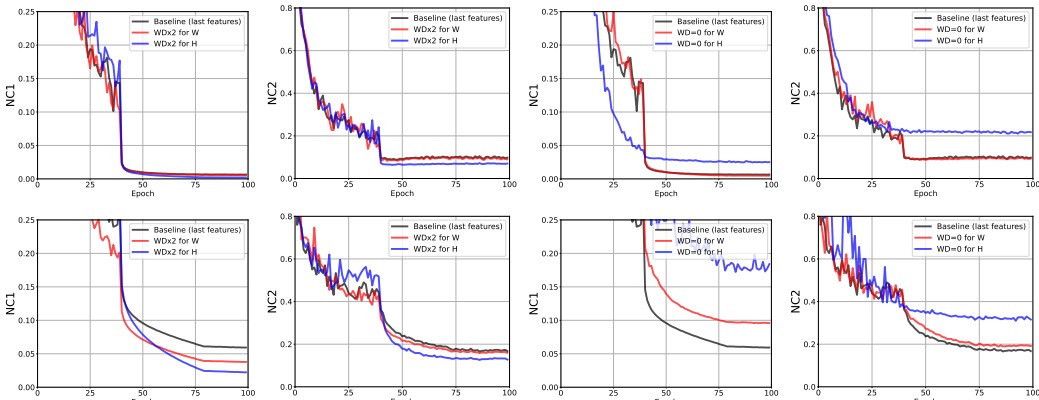

Figure 3: The effect of modifying the weight decay (WD) on NC metrics for ResNet18 trained on CIFAR-10. Top: MSE loss without bias; Bottom: CE loss with bias. Observe that modifying the WD in the feature mapping increases the deviation from the baseline more than modifying the WD of the last layer.

We consider the CIFAR-10 dataset and examine how modifying the regularization hyperparameters affects the NC behavior of the widely used ResNet18 (He et al., 2016a) compared to a baseline setting. Specifically, as a baseline hyperparameter setting, we consider one that is used in previous works (Papyan et al., 2020; Zhu et al., 2021): default PyTorch initialization of the weights, SGD optimizer with learning rate 0.05 that is divided by 10 every 40 epochs, momentum of 0.9, and WD of 5e-4 for all the network's parameters. The modifications include: 1) doubling the WD only for the last (FC) layer; 2) doubling the WD only for feature mapping (conv) layers; 3) zeroing the WD for the last layer; and 4) zeroing the WD for feature mapping layers.

Figure 3 presents the NC1 and NC2 metrics of the (deepest) features for: (Top) MSE loss with no bias in the FC layer (similar to the analyzed model); and (Bottom) CE loss with bias in the FC layer. In all the settings, we reach zero training error at the 40 epoch approximately. The empirical results show that modifying the WD in the feature mapping layers leads to curves with larger deviations from the baseline compared to modifying the last layer's WD, which is aligned with the theory established in Section 4 (i.e., the important role of $\lambda_H$ in attenuating the dominant intra-class perturbations). Reducing (zeroing) the WD in the feature mapping increases the distance from exact NC (i.e., from 0 value of the metrics), while increasing the WD decreases the gap from exact NC, as the theory predicts. The fact that sometimes (e.g., with CE loss) increasing the WD of the last layer can also decrease the gap from collapse hints that mitigating inter-class interference/correlation of features in practical deep learning settings is more significant for reaching NC than in our analysis that considers a near-collapse regime.[9] Yet, both the experiments and the theoretical study show that the regularization of the feature mapping has larger significance in approaching NC.

## 6 CONCLUSION

The features that are learned by training practical networks on real world datasets typically do not reach exact NC. In this paper, we addressed this issue by studying a model that can force the features to stay in the vicinity of a predefined features matrix. We analyzed it for the small vicinity case and established results that cannot be obtained by the previously studied (idealized) UFMs. We proved reduction in within-class variability of the optimized features compared to the input features (via analyzing gradient flow along the "central-path" of a UFM with minimal assumptions, unlike existing literature). We also presented an analysis of the model's minimizer in the near-collapse regime that provides insights on the effect of the regularization hyperparameters on the closeness to collapse, which correlate with the behavior in practical deep learning settings. We believe that our perturbation analysis approach, which is based on exploiting our knowledge on exactly collapsed minimizers of UFMs for studying non-collapse cases, can be applied to models other than the one considered in this paper, such as models with different loss functions and/or multiple levels of features and/or imbalanced data.

---

[9]In Appendix E.2, we demonstrate the role of $\lambda_W$ in mitigating inter-class interference of features, which is identified by our analysis, also empirically with practical DNNs.

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

# A    PROOF OF THEOREM 3.1

To prove Theorem 3.1, in addition to the within-class and between-class covariance matrices, let us define the total covariance matrix (across all classes) of the non-centered features

$$\tilde{\boldsymbol{\Sigma}}_T(\mathbf{H}) := \frac{1}{Kn} \sum_{k=1}^{K} \sum_{i=1}^{n} \mathbf{h}_{k,i} \mathbf{h}_{k,i}^{\top}.$$

For convenience we also define the non-centered between-class covariance matrix

$$\tilde{\boldsymbol{\Sigma}}_B(\mathbf{H}) := \frac{1}{K} \sum_{k=1}^{K} \overline{\mathbf{h}}_k \overline{\mathbf{h}}_k^{\top}.$$

We have the decomposition $\tilde{\boldsymbol{\Sigma}}_T(\mathbf{H}) = \boldsymbol{\Sigma}_W(\mathbf{H}) + \tilde{\boldsymbol{\Sigma}}_B(\mathbf{H})$.

Using $\mathbf{Y}\mathbf{H}^{\top} = (\mathbf{I}_K \otimes \mathbf{1}_n^{\top})\mathbf{H}^{\top} = n\overline{\mathbf{H}}^{\top}$ and $\tilde{\boldsymbol{\Sigma}}_T = \frac{1}{Kn}\mathbf{H}\mathbf{H}^{\top}$, we have that for each feature matrix $\mathbf{H}$, the optimal weight matrix $\mathbf{W}^*(\mathbf{H})$ is given by

$$\mathbf{W}^*(\mathbf{H}) = \frac{1}{K}\overline{\mathbf{H}}^{\top}(\frac{1}{Kn}\mathbf{H}\mathbf{H}^{\top} + \frac{\lambda_W}{K}\mathbf{I})^{-1} = \frac{1}{K}\overline{\mathbf{H}}^{\top}(\tilde{\boldsymbol{\Sigma}}_T + \frac{\lambda_W}{K}\mathbf{I})^{-1}.$$

Next, let us simplify the terms with $\mathbf{W}^*(\mathbf{H})$ in $\mathcal{L}(\mathbf{H})$:

$$\mathcal{L}(\mathbf{H}) := \frac{1}{2Kn}\|\mathbf{W}^*(\mathbf{H})\mathbf{H} - \mathbf{Y}\|_F^2 + \frac{\lambda_W}{2K}\|\mathbf{W}^*(\mathbf{H})\|_F^2 + \frac{\lambda_H}{2Kn}\|\mathbf{H}\|_F^2.$$

For the first term in $\mathcal{L}(\mathbf{H})$, observe that

$$\frac{1}{2Kn}\|\mathbf{W}^*(\mathbf{H})\mathbf{H} - \mathbf{Y}\|_F^2 = \frac{1}{2Kn}\mathrm{Tr}\left(\mathbf{W}^*(\mathbf{H})\mathbf{H}\mathbf{H}^{\top}\mathbf{W}^*(\mathbf{H})^{\top}\right) - \frac{1}{Kn}\mathrm{Tr}\left(\mathbf{W}^*(\mathbf{H})\mathbf{H}\mathbf{Y}^{\top}\right) + \frac{1}{2}$$

$$= \frac{1}{2K}\mathrm{Tr}\left((\tilde{\boldsymbol{\Sigma}}_T + \frac{\lambda_W}{K}\mathbf{I})^{-1}\tilde{\boldsymbol{\Sigma}}_T(\tilde{\boldsymbol{\Sigma}}_T + \frac{\lambda_W}{K}\mathbf{I})^{-1}\tilde{\boldsymbol{\Sigma}}_B\right) - \frac{1}{K}\mathrm{Tr}\left((\tilde{\boldsymbol{\Sigma}}_T + \frac{\lambda_W}{K}\mathbf{I})^{-1}\tilde{\boldsymbol{\Sigma}}_B\right) + \frac{1}{2}$$

$$= -\frac{\lambda_W}{2K^2}\mathrm{Tr}\left((\tilde{\boldsymbol{\Sigma}}_T + \frac{\lambda_W}{K}\mathbf{I})^{-2}\tilde{\boldsymbol{\Sigma}}_B\right) - \frac{1}{2K}\mathrm{Tr}\left((\tilde{\boldsymbol{\Sigma}}_T + \frac{\lambda_W}{K}\mathbf{I})^{-1}\tilde{\boldsymbol{\Sigma}}_B\right) + \frac{1}{2},$$

where in the second equality we used $\tilde{\boldsymbol{\Sigma}}_B = \frac{1}{K}\overline{\mathbf{H}}\overline{\mathbf{H}}^{\top}$, and in the last equality we used $(\tilde{\boldsymbol{\Sigma}}_T + \frac{\lambda_W}{K}\mathbf{I})^{-1}\tilde{\boldsymbol{\Sigma}}_T = \mathbf{I} - \frac{\lambda_W}{K}(\tilde{\boldsymbol{\Sigma}}_T + \frac{\lambda_W}{K}\mathbf{I})^{-1}$.

For the second term in $\mathcal{L}(\mathbf{H})$, observe that

$$\frac{\lambda_W}{2K}\|\mathbf{W}^*(\mathbf{H})\|_F^2 = \frac{\lambda_W}{2K}\mathrm{Tr}\left(\mathbf{W}^*(\mathbf{H})\mathbf{W}^*(\mathbf{H})^{\top}\right)$$

$$= \frac{\lambda_W}{2K^2}\mathrm{Tr}\left((\tilde{\boldsymbol{\Sigma}}_T + \frac{\lambda_W}{K}\mathbf{I})^{-2}\tilde{\boldsymbol{\Sigma}}_B\right).$$

Adding the two terms together,

$$\frac{1}{2Kn}\|\mathbf{W}^*(\mathbf{H})\mathbf{H} - \mathbf{Y}\|_F^2 + \frac{\lambda_W}{2K}\|\mathbf{W}^*(\mathbf{H})\|_F^2 = -\frac{1}{2K}\mathrm{Tr}\left((\tilde{\boldsymbol{\Sigma}}_T + \frac{\lambda_W}{K}\mathbf{I})^{-1}\tilde{\boldsymbol{\Sigma}}_B\right) + \frac{1}{2}$$

$$= \frac{1}{2K}\mathrm{Tr}\left((\tilde{\boldsymbol{\Sigma}}_T + \frac{\lambda_W}{K}\mathbf{I})^{-1}(\boldsymbol{\Sigma}_W + \frac{\lambda_W}{K}\mathbf{I})\right) - \frac{d-K}{2K},$$

where we used $(\tilde{\boldsymbol{\Sigma}}_T + \frac{\lambda_W}{K}\mathbf{I})^{-1}\tilde{\boldsymbol{\Sigma}}_B = \mathbf{I} - (\tilde{\boldsymbol{\Sigma}}_T + \frac{\lambda_W}{K}\mathbf{I})^{-1}(\boldsymbol{\Sigma}_W + \frac{\lambda_W}{K}\mathbf{I})$.

Finally, for the third term in $\mathcal{L}(\mathbf{H})$ we have

$$\frac{\lambda_H}{2Kn}\|\mathbf{H}\|_F^2 = \frac{\lambda_H}{2}\mathrm{Tr}\left(\tilde{\boldsymbol{\Sigma}}_T\right).$$

To conclude

$$\mathcal{L}(\mathbf{H}) = \frac{1}{2K}\mathrm{Tr}\left((\mathbf{\Sigma}_W + \frac{\lambda_W}{K}\mathbf{I})(\tilde{\mathbf{\Sigma}}_T + \frac{\lambda_W}{K}\mathbf{I})^{-1}\right) + \frac{\lambda_H}{2}\mathrm{Tr}\left(\tilde{\mathbf{\Sigma}}_T\right) - \frac{d-K}{2K}.$$

Next, we are going to analyze the traces of $\frac{d\mathbf{\Sigma}_B}{dt}$, $\frac{d\mathbf{\Sigma}_W}{dt}$, and $\frac{d\tilde{\mathbf{\Sigma}}_T}{dt}$, along the flow that is stated in Eq. 4, which is repeated here for the convenience of the reader:

$$\frac{d\mathbf{H}_t}{dt} = -Kn\nabla\mathcal{L}(\mathbf{H}_t).$$

In the following lemma, we state the required derivatives.

**Lemma A.1.** *Denote* $\mathbf{C}_B := \mathbf{\Sigma}_B(\tilde{\mathbf{\Sigma}}_T + \frac{\lambda_W}{K}\mathbf{I})^{-1}$, $\tilde{\mathbf{C}}_B := \tilde{\mathbf{\Sigma}}_B(\tilde{\mathbf{\Sigma}}_T + \frac{\lambda_W}{K}\mathbf{I})^{-1}$ *and* $\mathbf{C}_W := \mathbf{\Sigma}_W(\tilde{\mathbf{\Sigma}}_T + \frac{\lambda_W}{K}\mathbf{I})^{-1}$. *Along the gradient flow we have*

$$\frac{d\mathbf{\Sigma}_B}{dt} = \frac{1}{K}\left(\mathbf{C}_B(\mathbf{I} - \tilde{\mathbf{C}}_B) + (\mathbf{I} - \tilde{\mathbf{C}}_B^\top)\mathbf{C}_B^\top\right) - 2\lambda_H\mathbf{\Sigma}_B$$

$$\frac{d\mathbf{\Sigma}_W}{dt} = -\frac{1}{K}\left(\mathbf{C}_W\tilde{\mathbf{C}}_B + \tilde{\mathbf{C}}_B^\top\mathbf{C}_W^\top\right) - 2\lambda_H\mathbf{\Sigma}_W$$

$$\frac{d\tilde{\mathbf{\Sigma}}_T}{dt} = \frac{1}{K}\left((\mathbf{I} - \tilde{\mathbf{C}}_B - \mathbf{C}_W)\tilde{\mathbf{C}}_B + \tilde{\mathbf{C}}_B^\top(\mathbf{I} - \tilde{\mathbf{C}}_B^\top - \mathbf{C}_W^\top)\right) - 2\lambda_H\tilde{\mathbf{\Sigma}}_T$$

*Proof.* We use the notation $\partial_{kjl}$ to denote the derivative w.r.t. the $l$th entry of $\mathbf{h}_{k,j}$. Then

$$\partial_{kjl}\mathbf{\Sigma}_B = \frac{1}{Kn}(\mathbf{e}_l(\overline{\mathbf{h}}_k - \overline{\mathbf{h}}_G)^\top + (\overline{\mathbf{h}}_k - \overline{\mathbf{h}}_G)\mathbf{e}_l^\top),$$

$$\partial_{kjl}\mathbf{\Sigma}_W = \frac{1}{Kn}\left(\mathbf{e}_l(\mathbf{h}_{k,j} - \overline{\mathbf{h}}_k)^\top + (\mathbf{h}_{k,j} - \overline{\mathbf{h}}_k)\mathbf{e}_l^\top\right),$$

$$\partial_{kjl}\tilde{\mathbf{\Sigma}}_T = \frac{1}{Kn}(\mathbf{e}_l\mathbf{h}_{k,j}^\top + \mathbf{h}_{k,j}\mathbf{e}_l^\top),$$

where $\mathbf{e}_l \in \mathbf{R}^d$ is the one-hot vector whose $l$th entry is one (i.e., a standard basis vector). By the product rule,

$$\partial_{kjl}\mathcal{L}(\mathbf{H}) = \frac{1}{2K}\mathrm{Tr}\left((\partial_{kjl}\mathbf{\Sigma}_W)(\tilde{\mathbf{\Sigma}}_T + \frac{\lambda_W}{K}\mathbf{I})^{-1}\right) + \frac{1}{2K}\mathrm{Tr}\left((\mathbf{\Sigma}_W + \frac{\lambda_W}{K}\mathbf{I})\partial_{kjl}\left(\tilde{\mathbf{\Sigma}}_T + \frac{\lambda_W}{K}\mathbf{I}\right)^{-1}\right) + \frac{\lambda_H}{Kn}\mathbf{e}_l^\top\mathbf{h}_{k,j}$$

$$= \frac{1}{2K}\mathrm{Tr}\left((\partial_{kjl}\mathbf{\Sigma}_W)(\tilde{\mathbf{\Sigma}}_T + \frac{\lambda_W}{K}\mathbf{I})^{-1}\right)$$

$$- \frac{1}{2K}\mathrm{Tr}\left((\mathbf{\Sigma}_W + \frac{\lambda_W}{K}\mathbf{I})\left(\tilde{\mathbf{\Sigma}}_T + \frac{\lambda_W}{K}\mathbf{I}\right)^{-1}\partial_{kjl}\tilde{\mathbf{\Sigma}}_T\left(\tilde{\mathbf{\Sigma}}_T + \frac{\lambda_W}{K}\mathbf{I}\right)^{-1}\right) + \frac{\lambda_H}{Kn}\mathbf{e}_l^\top\mathbf{h}_{k,j}$$

$$= \frac{1}{K^2n}\left((\tilde{\mathbf{\Sigma}}_T + \frac{\lambda_W}{K}\mathbf{I})^{-1}(\mathbf{h}_{k,j} - \overline{\mathbf{h}}_k) - \left(\tilde{\mathbf{\Sigma}}_T + \frac{\lambda_W}{K}\mathbf{I}\right)^{-1}(\mathbf{\Sigma}_W + \frac{\lambda_W}{K}\mathbf{I})\left(\tilde{\mathbf{\Sigma}}_T + \frac{\lambda_W}{K}\mathbf{I}\right)^{-1}\mathbf{h}_{k,j} + \lambda_H K\mathbf{h}_{k,j}\right)^\top\mathbf{e}_l.$$

Therefore, the gradient of $\mathcal{L}$ is given by

$$\nabla\mathcal{L}(\mathbf{H}) = \tag{7}$$

$$\frac{1}{K^2n}\left((\tilde{\mathbf{\Sigma}}_T + \frac{\lambda_W}{K}\mathbf{I})^{-1}(\mathbf{H} - \overline{\mathbf{H}} \otimes \mathbf{1}_n^\top) - \left(\tilde{\mathbf{\Sigma}}_T + \frac{\lambda_W}{K}\mathbf{I}\right)^{-1}\left(\mathbf{\Sigma}_W + \frac{\lambda_W}{K}\mathbf{I}\right)\left(\tilde{\mathbf{\Sigma}}_T + \frac{\lambda_W}{K}\mathbf{I}\right)^{-1}\mathbf{H} + \lambda_H K\mathbf{H}\right).$$

Next, we compute how each covariance matrix updates along the flow. Let $\mathbf{\Sigma}_B(a,b) = \mathbf{e}_a^\top\mathbf{\Sigma}_B\mathbf{e}_b$ denote the $a,b$-th entry of $\mathbf{\Sigma}_B$. We further denote $\mathbf{C} := (\tilde{\mathbf{\Sigma}}_T + \frac{\lambda_W}{K}\mathbf{I})^{-1}$, $\mathbf{C}_B := \mathbf{\Sigma}_B(\tilde{\mathbf{\Sigma}}_T + \frac{\lambda_W}{K}\mathbf{I})^{-1}$, $\tilde{\mathbf{C}}_B := \tilde{\mathbf{\Sigma}}_B(\tilde{\mathbf{\Sigma}}_T + \frac{\lambda_W}{K}\mathbf{I})^{-1}$ $\mathbf{C}_W := \mathbf{\Sigma}_W(\tilde{\mathbf{\Sigma}}_T + \frac{\lambda_W}{K}\mathbf{I})^{-1}$ and write $\partial_{kjl}\mathcal{L}(\mathbf{H}) = \langle\mathbf{L}_{kj}, \mathbf{e}_l\rangle$, where

$$\mathbf{L}_{kj} = \frac{1}{K^2n}\left(\mathbf{C}(\mathbf{h}_{k,j} - \overline{\mathbf{h}}_k) - (\mathbf{I} - \tilde{\mathbf{C}}_B^\top)\mathbf{C}\mathbf{h}_{k,j} + \lambda_H K\mathbf{h}_{k,j}\right).$$

Using the chain rule, we have that

$$
\begin{aligned}
\frac{d\mathbf{\Sigma}_B(a,b)}{dt} &= \sum_{k,j,l} \partial_{kjl}\mathbf{\Sigma}_B(a,b)\frac{d\mathbf{h}_{k,j}[\ell]}{dt} = \sum_{k,j,l} \partial_{kjl}\mathbf{\Sigma}_B(a,b)(-Kn\partial_{kjl}\mathcal{L}(\mathbf{H})) \\
&= \sum_{k,j}\sum_{l} -\left(\langle \mathbf{e}_a, \mathbf{e}_l\rangle\langle \mathbf{e}_b, \overline{\mathbf{h}}_k - \overline{\mathbf{h}}_G\rangle + \langle \mathbf{e}_a, \overline{\mathbf{h}}_k - \overline{\mathbf{h}}_G\rangle\langle \mathbf{e}_l, \mathbf{e}_b\rangle\right)\langle \mathbf{e}_l, \mathbf{L}_{kjl}\rangle \\
&= \sum_{k,j} -\left(\langle \mathbf{e}_a, \mathbf{L}_{kj}\rangle\langle \mathbf{e}_b, \overline{\mathbf{h}}_k - \overline{\mathbf{h}}_G\rangle + \langle \mathbf{e}_a, \overline{\mathbf{h}}_k - \overline{\mathbf{h}}_G\rangle\langle \mathbf{L}_{kj}, \mathbf{e}_b\rangle\right) \\
&= \mathbf{e}_a^T\left(\sum_{k,j} -\mathbf{L}_{k,j}(\overline{\mathbf{h}}_k - \overline{\mathbf{h}}_G)^\top - (\overline{\mathbf{h}}_k - \overline{\mathbf{h}}_G)\mathbf{L}_{k,j}^\top\right)\mathbf{e}_b \\
&= \frac{1}{K}\mathbf{e}_a^T\left(\mathbf{C}_B(\mathbf{I} - \tilde{\mathbf{C}}_B) + (\mathbf{I} - \tilde{\mathbf{C}}_B^\top)\mathbf{C}_B^\top\right)\mathbf{e}_b - 2\lambda_H\mathbf{e}_a^\top\mathbf{\Sigma}_B\mathbf{e}_b
\end{aligned}
$$

Similar computation yields

$$
\begin{aligned}
\frac{d\mathbf{\Sigma}_W(a,b)}{dt} &= -\frac{1}{K}\mathbf{e}_a^\top\left(\mathbf{C}_W\tilde{\mathbf{C}}_B + \tilde{\mathbf{C}}_B^\top\mathbf{C}_W^\top\right)\mathbf{e}_b - 2\lambda_H\mathbf{e}_a^\top\mathbf{\Sigma}_W\mathbf{e}_b \\
\frac{d\tilde{\mathbf{\Sigma}}_T(a,b)}{dt} &= \frac{1}{K}\mathbf{e}_a^T\left((\mathbf{I} - \tilde{\mathbf{C}}_B - \mathbf{C}_W)\tilde{\mathbf{C}}_B + \tilde{\mathbf{C}}_B^\top(\mathbf{I} - \tilde{\mathbf{C}}_B^\top - \mathbf{C}_W^\top)\right)\mathbf{e}_b - 2\lambda_H\mathbf{e}_a^\top\tilde{\mathbf{\Sigma}}_T\mathbf{e}_b
\end{aligned}
$$

$\square$

Let $T_B : t \mapsto e^{2\lambda_H t}\mathrm{Tr}(\mathbf{\Sigma}_B)$ and $T_W : t \mapsto e^{2\lambda_H t}\mathrm{Tr}(\mathbf{\Sigma}_W)$. The above lemma suggests that $T_B$ strictly increases along the flow, while $T_W$ decreases. Indeed,

$$
\begin{aligned}
\frac{dT_W}{dt} &= e^{2\lambda_H t}\left(\frac{d\,\mathrm{Tr}(\mathbf{\Sigma}_W)}{dt} + 2\lambda_H\mathrm{Tr}(\mathbf{\Sigma}_W)\right) \\
&= -\frac{2}{K}e^{2\lambda_H t}\mathrm{Tr}(\mathbf{C}_W\tilde{\mathbf{C}}_B) \\
&= -e^{2\lambda_H t}\frac{2}{K}\mathrm{Tr}(\mathbf{\Sigma}_W(\tilde{\mathbf{\Sigma}}_T + \frac{\lambda_W}{K}\mathbf{I})^{-1}\tilde{\mathbf{\Sigma}}_B(\tilde{\mathbf{\Sigma}}_T + \frac{\lambda_W}{K}\mathbf{I})^{-1}) \le 0,
\end{aligned}
$$

The last inequality holds because the trace of the product of two positive semidefinite matrices is always non-negative (e.g. by Von-Neumann's trace inequality). Similarly

$$
\begin{aligned}
\frac{dT_B}{dt} &= \frac{2}{K}e^{2\lambda_H t}\mathrm{Tr}(\mathbf{C}_B(\mathbf{I} - \tilde{\mathbf{C}}_B)) \\
&= \frac{2}{K}e^{2\lambda_H t}\mathrm{Tr}(\mathbf{\Sigma}_B(\tilde{\mathbf{\Sigma}}_T + \frac{\lambda_W}{K}\mathbf{I})^{-1}(\mathbf{I} - \tilde{\mathbf{\Sigma}}_B(\tilde{\mathbf{\Sigma}}_T + \frac{\lambda_W}{K}\mathbf{I})^{-1})) \\
&= \frac{2}{K}e^{2\lambda_H t}\mathrm{Tr}(\mathbf{\Sigma}_B(\tilde{\mathbf{\Sigma}}_T + \frac{\lambda_W}{K}\mathbf{I})^{-1}(\mathbf{\Sigma}_W + \frac{\lambda_W}{K}\mathbf{I})(\tilde{\mathbf{\Sigma}}_T + \frac{\lambda_W}{K}\mathbf{I})^{-1}) \\
&= \frac{2}{K}e^{2\lambda_H t}\left(\mathrm{Tr}(\mathbf{\Sigma}_B(\tilde{\mathbf{\Sigma}}_T + \frac{\lambda_W}{K}\mathbf{I})^{-1}\mathbf{\Sigma}_W(\tilde{\mathbf{\Sigma}}_T + \frac{\lambda_W}{K}\mathbf{I})^{-1}) + \frac{\lambda_W}{K}\mathrm{Tr}(\mathbf{\Sigma}_B(\tilde{\mathbf{\Sigma}}_T + \frac{\lambda_W}{K}\mathbf{I})^{-2})\right) \\
&\ge \frac{2\lambda_W}{K^2}e^{2\lambda_H t}\mathrm{Tr}(\mathbf{\Sigma}_B(\tilde{\mathbf{\Sigma}}_T + \frac{\lambda_W}{K}\mathbf{I})^{-2}) > 0,
\end{aligned}
$$

where the strict inequality again comes from Von-Neumann trace inequality, which ensures that the trace of product of a positive definite matrix and a non-zero positive semidefinite matrix is positive.

Since $\widetilde{NC}_1 = T_W/T_B$, the above computation also shows that $\widetilde{NC}_1$ has to strictly decrease along the flow.

## B   PROOF OF COROLLARY 3.2

Recall that the minimizer $\mathbf{H}_{1/\beta}$ satisfies the first order equation

$$\mathbf{H}_{1/\beta} - \mathbf{H}_0 = -\frac{Kn}{\beta}\nabla\mathcal{L}(\mathbf{H}_{1/\beta}). \tag{8}$$

We first show that $\mathbf{H}_{1/\beta} \to \mathbf{H}_0$ as $\beta \to \infty$. The following lemma would be helpful.

**Lemma B.1.** *There exists a constant $M > 0$ independent of $\mathbf{H}$, such that*

$$\|\nabla\mathcal{L}(\mathbf{H})\|_F \leq M\|\mathbf{H}\|_F,$$

*for any $\mathbf{H} \in \mathbb{R}^{d \times Kn}$.*

*Proof.* We bound each term in the expression of $\nabla\mathcal{L}$ equation Eq. 7 individually. For the first term we have

$$\|(\tilde{\boldsymbol{\Sigma}}_T + \frac{\lambda_W}{K}\mathbf{I})^{-1}(\mathbf{H} - \overline{\mathbf{H}} \otimes \mathbf{1}_n^\top)\|_F \leq \|(\tilde{\boldsymbol{\Sigma}}_T + \frac{\lambda_W}{K}\mathbf{I})^{-1}\|_{op}\|(\mathbf{H} - \overline{\mathbf{H}} \otimes \mathbf{1}_n^\top)\|_F$$
$$\leq \frac{K}{\lambda_W}\|(\mathbf{H} - \overline{\mathbf{H}} \otimes \mathbf{1}_n^\top)\|_F \leq \frac{2K}{\lambda_W}\|\mathbf{H}\|_F,$$

where $\|\cdot\|_{op}$ denotes the operator norm and the second inequality is due to the fact that each eigenvalue of $(\tilde{\boldsymbol{\Sigma}}_T + \frac{\lambda_W}{K}\mathbf{I})^{-1}$ is no bigger than $\frac{K}{\lambda_W}$. Similarly,

$$\left\|\left(\tilde{\boldsymbol{\Sigma}}_T + \frac{\lambda_W}{K}\mathbf{I}\right)^{-1}\left(\boldsymbol{\Sigma}_W + \frac{\lambda_W}{K}\mathbf{I}\right)\left(\tilde{\boldsymbol{\Sigma}}_T + \frac{\lambda_W}{K}\mathbf{I}\right)^{-1}\mathbf{H}\right\|_F$$
$$\leq \frac{K}{\lambda_W}\left\|\left(\tilde{\boldsymbol{\Sigma}}_T + \frac{\lambda_W}{K}\mathbf{I}\right)^{-\frac{1}{2}}\left(\boldsymbol{\Sigma}_W + \frac{\lambda_W}{K}\mathbf{I}\right)\left(\tilde{\boldsymbol{\Sigma}}_T + \frac{\lambda_W}{K}\mathbf{I}\right)^{-\frac{1}{2}}\right\|_{op}\|\mathbf{H}\|_F,$$

where in the last inequality we used $\|(\tilde{\boldsymbol{\Sigma}}_T + \frac{\lambda_W}{K}\mathbf{I})^{-1/2}\|_{op} \leq \sqrt{K/\lambda_W}$ since every eigenvalue of $(\tilde{\boldsymbol{\Sigma}}_T + \frac{\lambda_W}{K}\mathbf{I})^{-1/2}$ is bounded by $\sqrt{K/\lambda_W}$. Denote $\mathbf{A} = \left(\boldsymbol{\Sigma}_W + \frac{\lambda_W}{K}\mathbf{I}\right)$, $\mathbf{B} = \left(\tilde{\boldsymbol{\Sigma}}_T + \frac{\lambda_W}{K}\mathbf{I}\right)$ and use $\mathbf{A} + \tilde{\boldsymbol{\Sigma}}_B = \mathbf{B}$, we have

$$\|\mathbf{B}^{-1/2}\mathbf{A}\mathbf{B}^{-1/2}\|_{op} = \|(\mathbf{B}^{-1/2}\mathbf{A}^{1/2})(\mathbf{B}^{-1/2}\mathbf{A}^{1/2})^\top\|_{op}$$
$$= \|(\mathbf{B}^{-1/2}\mathbf{A}^{1/2})^\top(\mathbf{B}^{-1/2}\mathbf{A}^{1/2})\|_{op}$$
$$= \|\mathbf{A}^{1/2}\mathbf{B}^{-1}\mathbf{A}^{1/2}\|_{op}$$
$$= \|(\mathbf{A}^{-1/2}(\mathbf{A} + \tilde{\boldsymbol{\Sigma}}_B)\mathbf{A}^{-1/2})^{-1}\|_{op}$$
$$= \|(\mathbf{I} + \mathbf{A}^{-1/2}\tilde{\boldsymbol{\Sigma}}_B\mathbf{A}^{-1/2})^{-1}\|_{op} \leq 1.$$

Combining the above bounds together, we have obtained for any $\mathbf{H} \in \mathbb{R}^{d \times Kn}$,

$$\|\nabla\mathcal{L}(\mathbf{H})\|_F \leq \frac{1}{Kn}\left(\frac{3}{\lambda_W} + \lambda_H\right)\|\mathbf{H}\|_F.$$

$\square$

Next, we combine the lemma and the stationary equation Eq. 8 to get

$$\|\mathbf{H}_{1/\beta} - \mathbf{H}_0\|_F \leq \frac{nKM}{\beta}\|\mathbf{H}_{1/\beta}\|_F \leq \frac{nKM}{\beta}\|\mathbf{H}_{1/\beta} - \mathbf{H}_0\|_F + \frac{nKM}{\beta}\|\mathbf{H}_0\|_F.$$

Rearranging, we have the bound

$$\|\mathbf{H}_{1/\beta} - \mathbf{H}_0\|_F \leq \left(\frac{\beta}{nKM} - 1\right)^{-1}\|\mathbf{H}_0\|_F.$$

This implies that $\mathbf{H}_{1/\beta} \to \mathbf{H}_0$ as $\beta \to \infty$. Combined with the continuity of $\nabla\mathcal{L}(\cdot)$ and the first order equation Eq. 8, this further implies

$$\lim_{\beta \to \infty} \frac{\mathbf{H}_{1/\beta} - \mathbf{H}_0}{1/\beta} = -Kn\nabla\mathcal{L}(\mathbf{H}_0).$$

Now, by chain rule,

$$\begin{aligned} \lim_{\beta \to \infty} \frac{\widetilde{NC}_1(\mathbf{H}_{1/\beta}) - \widetilde{NC}_1(\mathbf{H}_0)}{1/\beta} &= \langle \nabla_H \widetilde{NC}_1(\mathbf{H}_0), \lim_{\beta \to \infty} \frac{\mathbf{H}_{1/\beta} - \mathbf{H}_0}{1/\beta} \rangle \\ &= \langle \nabla_H \widetilde{NC}_1(\mathbf{H}_0), -Kn\nabla\mathcal{L}(\mathbf{H}_0) \rangle \\ &= \frac{d}{dt}\bigg|_{t=0} \widetilde{NC}_1(\mathbf{H}_t). \end{aligned}$$

In the last line, $\mathbf{H}_t$ denotes the gradient flow iterate defined in Eq. 4. By (the proof of) Theorem 3.1, when $\mathbf{H}_0$ is non-collapsed,

$$\frac{d}{dt}\bigg|_{t=0} \widetilde{NC}_1(\mathbf{H}_t) < 0$$

must hold. This further implies that there exists some constant $C = C(\mathbf{H}_0) > 0$ such that for $\beta > C$ we have that $\frac{\widetilde{NC}_1(\mathbf{H}_{1/\beta}) - \widetilde{NC}_1(\mathbf{H}_0)}{1/\beta} < 0$.

## C  PROOF OF THEOREM 4.1

Our proof is essentially a perturbation analysis approach that exploits the fact that each of the minimizers is a stationary point of its associated objective function. Namely, the minimizer of the perturbed problem $f(\mathbf{W}, \mathbf{H}; \tilde{\mathbf{H}}_0)$, i.e., $(\tilde{\mathbf{W}}^*, \tilde{\mathbf{H}}^*)$, obeys that $\nabla f(\tilde{\mathbf{W}}^*, \tilde{\mathbf{H}}^*; \tilde{\mathbf{H}}_0) = \begin{bmatrix} \nabla_H f(\tilde{\mathbf{W}}^*, \tilde{\mathbf{H}}^*; \tilde{\mathbf{H}}_0) \\ \nabla_W f(\tilde{\mathbf{W}}^*, \tilde{\mathbf{H}}^*; \tilde{\mathbf{H}}_0) \end{bmatrix} = \mathbf{0}$, and the minimizer of the unperturbed problem, i.e., $(\mathbf{W}^*, \mathbf{H}^*)$ where for brevity we omit the 'ˆ' symbol, obeys that $\nabla f(\mathbf{W}^*, \mathbf{H}^*; \mathbf{H}_0) = \begin{bmatrix} \nabla_H f(\mathbf{W}^*, \mathbf{H}^*; \mathbf{H}_0) \\ \nabla_W f(\mathbf{W}^*, \mathbf{H}^*; \mathbf{H}_0) \end{bmatrix} = \mathbf{0}$.

We use these properties in the following first order Taylor approximation of $\nabla f(\tilde{\mathbf{W}}^*, \tilde{\mathbf{H}}^*; \tilde{\mathbf{H}}_0)$ around $(\mathbf{W}^*, \mathbf{H}^*; \mathbf{H}_0)$ (with accuracy of $O(\|\delta \mathbf{H}\|^2, \|\delta \mathbf{W}\|^2, \|\delta \mathbf{H}_0\|^2)$) that is given by

$$
\begin{bmatrix} \nabla_H f(\tilde{\mathbf{W}}^*, \tilde{\mathbf{H}}^*; \tilde{\mathbf{H}}_0) \\ \nabla_W f(\tilde{\mathbf{W}}^*, \tilde{\mathbf{H}}^*; \tilde{\mathbf{H}}_0) \end{bmatrix} \approx \begin{bmatrix} \nabla_H f(\mathbf{W}^*, \mathbf{H}^*; \mathbf{H}_0) \\ \nabla_W f(\mathbf{W}^*, \mathbf{H}^*; \mathbf{H}_0) \end{bmatrix}
\tag{9}
$$
$$
+ \begin{bmatrix} \nabla_H^\top \nabla_H f(\mathbf{W}^*, \mathbf{H}^*; \mathbf{H}_0) & \nabla_W^\top \nabla_H f(\mathbf{W}^*, \mathbf{H}^*; \mathbf{H}_0) \\ \nabla_H^\top \nabla_W f(\mathbf{W}^*, \mathbf{H}^*; \mathbf{H}_0) & \nabla_W^\top \nabla_W f(\mathbf{W}^*, \mathbf{H}^*; \mathbf{H}_0) \end{bmatrix} \begin{bmatrix} \text{vec}(\delta \mathbf{H}) \\ \text{vec}(\delta \mathbf{W}) \end{bmatrix} + \begin{bmatrix} \nabla_{H_0}^\top \nabla_H f(\mathbf{W}^*, \mathbf{H}^*; \mathbf{H}_0) \\ \nabla_{H_0}^\top \nabla_W f(\mathbf{W}^*, \mathbf{H}^*; \mathbf{H}_0) \end{bmatrix} \text{vec}(\delta \mathbf{H}_0).
$$

Recall that $\delta \mathbf{H} := \tilde{\mathbf{H}}^* - \mathbf{H}^*$, $\delta \mathbf{W} := \tilde{\mathbf{W}}^* - \mathbf{W}^*$, and $\delta \mathbf{H}_0 = \tilde{\mathbf{H}}_0 - \mathbf{H}_0$. Since the two terms in the first line of Eq. 9 vanish, we get that

$$
\begin{bmatrix} \text{vec}(\delta \mathbf{H}) \\ \text{vec}(\delta \mathbf{W}) \end{bmatrix} \approx - \begin{bmatrix} \nabla_H^\top \nabla_H f & \nabla_W^\top \nabla_H f \\ \nabla_H^\top \nabla_W f & \nabla_W^\top \nabla_W f \end{bmatrix}^{-1} \begin{bmatrix} \nabla_{H_0}^\top \nabla_H f \\ \nabla_{H_0}^\top \nabla_W f \end{bmatrix} \text{vec}(\delta \mathbf{H}_0),
\tag{10}
$$

where all the derivatives are evaluated at $(\mathbf{W}^*, \mathbf{H}^*; \mathbf{H}_0)$, which is omitted in order to simplify the presentation. As shown below, in our setting the matrix that is inverted is indeed nonsingular.

We turn now to compute the derivatives. Let us denote $\mathbf{h} := \text{vec}(\mathbf{H})$, $\mathbf{w} := \text{vec}(\mathbf{W})$, and $\mathbf{y} := \text{vec}(\mathbf{Y})$. Observe that from well known identities on the Kronecker product and the vectorization operation we have

$$
\frac{1}{2Kn} \|\mathbf{W}\mathbf{H} - \mathbf{Y}\|_F^2 = \frac{1}{2Kn} \|(\mathbf{I}_{kn} \otimes \mathbf{W})\mathbf{h} - \mathbf{y}\|_2^2 = \frac{1}{2Kn} \|(\mathbf{H}^\top \otimes \mathbf{I}_K)\mathbf{w} - \mathbf{y}\|_2^2.
$$

Therefore, the first order derivatives are given by

$$
\nabla_H f(\mathbf{W}, \mathbf{H}; \mathbf{H}_0) = \frac{1}{Kn}(\mathbf{I}_{kn} \otimes \mathbf{W}^\top)((\mathbf{I}_{kn} \otimes \mathbf{W})\mathbf{h} - \mathbf{y}) + \frac{\lambda_H}{Kn}\mathbf{h} + \frac{\beta}{Kn}(\mathbf{h} - \text{vec}(\mathbf{H}_0)), \tag{11}
$$
$$
\nabla_W f(\mathbf{W}, \mathbf{H}; \mathbf{H}_0) = \frac{1}{Kn}(\mathbf{H} \otimes \mathbf{I}_K)((\mathbf{H}^\top \otimes \mathbf{I}_K)\mathbf{w} - \mathbf{y}) + \frac{\lambda_W}{K}\mathbf{w}.
$$

Hence,

$$
\nabla_{H_0}^\top \nabla_H f = -\frac{\beta}{Kn}\mathbf{I}_{dnK},
$$
$$
\nabla_{H_0}^\top \nabla_W f = \mathbf{0}_{Kd \times dnK}.
$$

Plugging these expressions in Eq. 10 and using blockwise matrix inversion gives

$$
\text{vec}(\delta \mathbf{H}) \approx \frac{\beta}{Kn} \left( \nabla_H^\top \nabla_H f - \nabla_W^\top \nabla_H f (\nabla_W^\top \nabla_W f)^{-1} \nabla_H^\top \nabla_W f \right)^{-1} \text{vec}(\delta \mathbf{H}_0),
$$

$$
\text{vec}(\delta \mathbf{W}) \approx -\frac{\beta}{Kn} (\nabla_W^\top \nabla_W f)^{-1} \nabla_H^\top \nabla_W f \left( \nabla_H^\top \nabla_H f - \nabla_W^\top \nabla_H f (\nabla_W^\top \nabla_W f)^{-1} \nabla_H^\top \nabla_W f \right)^{-1} \text{vec}(\delta \mathbf{H}_0),
$$

which are stated in the theorem, where all the derivatives are evaluated at the point $(\mathbf{W}^*, \mathbf{H}^*; \mathbf{H}_0)$.

Let us state the second order derivatives that appear above. First, one can observe that

$$
\nabla_H^\top \nabla_H f(\mathbf{W}^*, \mathbf{H}^*; \mathbf{H}_0) = \frac{1}{Kn}\mathbf{I}_{nK} \otimes \mathbf{W}^{*\top}\mathbf{W}^* + \frac{\lambda_H}{Kn}\mathbf{I}_{dnK} + \frac{\beta}{Kn}\mathbf{I}_{dnK},
$$
$$
\nabla_W^\top \nabla_W f(\mathbf{W}^*, \mathbf{H}^*; \mathbf{H}_0) = \frac{1}{Kn}\mathbf{H}\mathbf{H}^{*\top} \otimes \mathbf{I}_K + \frac{\lambda_W}{K}\mathbf{I}_{Kd}.
$$

As for the mixed partial derivative, applying $\nabla_W^\top$ on Eq. 11, we get

$$
\begin{aligned}
\nabla_W^\top \nabla_H f &= \frac{\partial}{\partial \mathbf{w}} \nabla_H f = \frac{1}{Kn} \frac{\partial}{\partial \mathbf{w}} \left( (\mathbf{I}_{kn} \otimes \mathbf{W}^\top) \mathbf{r} \right) + \frac{1}{Kn} (\mathbf{I}_{kn} \otimes \mathbf{W}^\top) \frac{\partial}{\partial \mathbf{w}} ((\mathbf{I}_{kn} \otimes \mathbf{W}) \mathbf{h} - \mathbf{y}) \\
&= \frac{1}{Kn} \frac{\partial}{\partial \mathbf{w}} \left( (\mathbf{I}_{kn} \otimes \mathbf{W}^\top) \mathbf{r} \right) + \frac{1}{Kn} (\mathbf{I}_{kn} \otimes \mathbf{W}^\top) \frac{\partial}{\partial \mathbf{w}} ((\mathbf{H}^\top \otimes \mathbf{I}_K) \mathbf{w} - \mathbf{y}) \\
&= \frac{1}{Kn} \mathbf{E}(\mathbf{W}, \mathbf{H}) + \frac{1}{Kn} (\mathbf{I}_{kn} \otimes \mathbf{W}^\top)(\mathbf{H}^\top \otimes \mathbf{I}_K) \\
&= \frac{1}{Kn} \mathbf{E}(\mathbf{W}, \mathbf{H}) + \frac{1}{Kn} (\mathbf{H}^\top \otimes \mathbf{W}^\top)
\end{aligned}
$$

where $\mathbf{r} := \mathrm{vec}(\mathbf{WH} - \mathbf{Y})$ but treated as independent of $\mathbf{w}$ due to the product rule, and $\mathbf{E}(\mathbf{W}, \mathbf{H}) := \frac{\partial}{\partial \mathbf{w}} \left( (\mathbf{I}_{kn} \otimes \mathbf{W}^\top) \mathbf{r} \right)$. Denoting $w_{k,i} := W[k,i]$, we have that

$$
\begin{aligned}
\frac{\partial}{\partial w_{k,i}} \left( (\mathbf{I}_{kn} \otimes \mathbf{W}^\top) \mathbf{r} \right) &= \left( (\mathbf{I}_{kn} \otimes \frac{\partial}{\partial w_{k,i}} \mathbf{W}^\top) \mathbf{r} \right) = \left( (\mathbf{I}_{kn} \otimes \mathbf{e}_{d,i} \mathbf{e}_{K,k}^\top) \mathbf{r} \right) \\
&= \mathrm{vec}(\mathbf{e}_{d,i} \mathbf{e}_{K,k}^\top (\mathbf{WH} - \mathbf{Y})),
\end{aligned}
$$

where $\mathbf{e}_{d,i}$ is the standard vector in $\mathbb{R}^d$ with 1 in its $i$th entry (similar definition stands for $\mathbf{e}_{K,k}$).

Therefore,

$$
\begin{aligned}
\nabla_H^\top \nabla_W f(\mathbf{W}^*, \mathbf{H}^*; \mathbf{H}_0) &= \frac{1}{Kn} \mathbf{E}^{*\top} + \frac{1}{Kn} (\mathbf{H}^* \otimes \mathbf{W}^*), \\
\nabla_W^\top \nabla_H f(\mathbf{W}^*, \mathbf{H}^*; \mathbf{H}_0) &= \frac{1}{Kn} \mathbf{E}^* + \frac{1}{Kn} (\mathbf{H}^{*\top} \otimes \mathbf{W}^{*\top}),
\end{aligned}
$$

where $\mathbf{E}^* = \mathbf{E}(\mathbf{W}^*, \mathbf{H}^*)$ and $\mathbf{E}(\mathbf{W}, \mathbf{H}) \in \mathbb{R}^{dnK \times Kd}$ is given by

$\mathbf{E}(\mathbf{W}, \mathbf{H}) :=$
$\big[ \mathrm{vec}(\mathbf{e}_{d,1} \mathbf{e}_{K,1}^\top (\mathbf{WH} - \mathbf{Y})), ..., \mathrm{vec}(\mathbf{e}_{d,1} \mathbf{e}_{K,K}^\top (\mathbf{WH} - \mathbf{Y})), \mathrm{vec}(\mathbf{e}_{d,2} \mathbf{e}_{K,1}^\top (\mathbf{WH} - \mathbf{Y})), ...$
$\qquad\qquad\qquad\qquad\qquad\qquad\qquad ..., \mathrm{vec}(\mathbf{e}_{d,d} \mathbf{e}_{K,K}^\top (\mathbf{WH} - \mathbf{Y})) \big].$

We focus now on the effect the deviation $\delta \mathbf{H}_0 = \tilde{\mathbf{H}}_0 - \mathbf{H}_0$ on the feature learning $\delta \mathbf{H} = \tilde{\mathbf{H}}^* - \mathbf{H}^*$. This requires inverting the $dnK \times dnK$ matrix that links $\delta \mathbf{H}$ and $\delta \mathbf{H}_0$, which is quite challenging. Yet, from the derivatives that are stated above we observe the following

$$
\begin{aligned}
\mathrm{vec}(\delta \mathbf{H}) &\approx \frac{\beta}{Kn} \left( \nabla_H^\top \nabla_H f - \nabla_W^\top \nabla_H f (\nabla_W^\top \nabla_W f)^{-1} \nabla_H^\top \nabla_W f \right)^{-1} \mathrm{vec}(\delta \mathbf{H}_0) \\
&= \left( \mathbf{I}_{dnK} + \frac{\lambda_H}{\beta} \mathbf{I}_{dnK} + \frac{1}{\beta} \mathbf{I}_{nK} \otimes \mathbf{W}^{*\top} \mathbf{W}^* - \frac{Kn}{\beta} \nabla_W^\top \nabla_H f (\nabla_W^\top \nabla_W f)^{-1} \nabla_H^\top \nabla_W f \right)^{-1} \mathrm{vec}(\delta \mathbf{H}_0) \\
&= \left( \mathbf{I}_{dnK} + \frac{\lambda_H}{\beta} \mathbf{I}_{dnK} + \frac{1}{\beta} \mathbf{I}_{nK} \otimes \mathbf{W}^{*\top} \mathbf{W}^* - \frac{1}{\beta} \mathbf{Z}^* \right)^{-1} \mathrm{vec}(\delta \mathbf{H}_0)
\end{aligned}
$$

where

$$
\mathbf{Z}^* := (\mathbf{E}^{*\top} + \mathbf{H}^* \otimes \mathbf{W}^*)^\top (\mathbf{H}^* \mathbf{H}^{*\top} \otimes \mathbf{I}_K + n\lambda_W \mathbf{I}_{dK})^{-1} (\mathbf{E}^{*\top} + \mathbf{H}^* \otimes \mathbf{W}^*).
$$

Therefore, under the assumption of $\beta \gg \max\{1, \lambda_H\}$, which is associated with a restrictive link between $\mathbf{H}_0$ and $\mathbf{H}$, we can use the first-order truncated Neumann series to approximate the matrix inversion (with accuracy of $O(\beta^{-2})$) that is stated in Eq. 5 and repeated here for the convenience of the reader:

$$
\mathrm{vec}(\delta \mathbf{H}) \approx \left( \mathbf{I}_{dnK} - \frac{\lambda_H}{\beta} \mathbf{I}_{dnK} - \frac{1}{\beta} \mathbf{I}_{nK} \otimes \mathbf{W}^{*\top} \mathbf{W}^* + \frac{1}{\beta} \mathbf{Z}^* \right) \mathrm{vec}(\delta \mathbf{H}_0).
$$

# D   PROOF OF THEOREM 4.2

In this section we compute the entire spectrum (singular values) for the diagonal blocks ("intra-class blocks") and the off-diagonal blocks ("inter-class blocks") of the block matrix in Eq. 6. To keep the main body of the paper concise, we present in the statement of Theorem 4.2 only the results for $\sigma_{max}(\mathbf{F}_{k,k})$ and $\sigma_{min}(\mathbf{F}_{k,k})$ of the full rank matrix $\mathbf{F}_{k,k}$, as well as $\sigma_{max}(\mathbf{F}_{k,\tilde{k}})$ of the rank-1 matrix $\mathbf{F}_{k,\tilde{k}}$ ($\tilde{k} \neq k$).

Recall that we consider the (non-degenerate) setting $c := \sqrt{\lambda_H \lambda_W} < 1$. Therefore, when $\mathbf{H}_0 = \mathbf{H}^*$ is a minimizer of Eq. 1 (associated with $\mathbf{W}^*$), from Corollary 2.2 we have that $(\mathbf{W}^*, \mathbf{H}^*)$ the minimizer of $f(\mathbf{W}, \mathbf{H}; \mathbf{H}_0)$ is also orthogonally collapsed and characterized by Theorem 2.1 with $\lambda_W$ and $\lambda_H$ (independent of $K, n, d$). That is, $\mathbf{H}^* = \overline{\mathbf{H}} \otimes \mathbf{1}_n^\top$ and $\mathbf{W}^* \overline{\mathbf{H}} \propto \overline{\mathbf{H}}^\top \overline{\mathbf{H}} \propto \mathbf{W}^* \mathbf{W}^{*\top} \propto \mathbf{I}_K$. We also have the following results for the spectral norm of $\overline{\mathbf{H}}$ and $\mathbf{W}^*$, that we denote by $\sigma_{\overline{H}}$ and $\sigma_W$ respectively:

$$\sigma_{\overline{H}}^2 = (1-c)\sqrt{\frac{\lambda_W}{\lambda_H}} = \sqrt{\frac{\lambda_W}{\lambda_H}} - \lambda_W,$$

$$\sigma_W^2 = (1-c)\sqrt{\frac{\lambda_H}{\lambda_W}} = \sqrt{\frac{\lambda_H}{\lambda_W}} - \lambda_H.$$

Observe that these expressions do not depend on the number of samples $K, n, d$. Note also that $\sigma_{\overline{H}}^2 \sigma_W^2 = (1 - \sqrt{\lambda_H \lambda_W})^2 = (1-c)^2 < 1$.

We remind the reader that $\text{vec}(\delta \mathbf{H}) \approx \mathbf{F} \text{vec}(\delta \mathbf{H}_0)$, for

$$\mathbf{F} = \mathbf{I}_{dnK} - \frac{\lambda_H}{\beta} \mathbf{I}_{dnK} - \frac{1}{\beta} \mathbf{I}_{nK} \otimes \mathbf{W}^{*\top} \mathbf{W}^* + \frac{1}{\beta} \mathbf{Z}^*,$$

where

$$\mathbf{Z}^* := (\mathbf{E}^{*\top} + \mathbf{H}^* \otimes \mathbf{W}^*)^\top (\mathbf{H}\mathbf{H}^{*\top} \otimes \mathbf{I}_K + n\lambda_W \mathbf{I}_{dK})^{-1} (\mathbf{E}^{*\top} + \mathbf{H}^* \otimes \mathbf{W}^*),$$

and $\mathbf{E}^* = \mathbf{E}(\mathbf{W}^*, \mathbf{H}^*)$ and $\mathbf{E}(\mathbf{W}, \mathbf{H}) \in \mathbb{R}^{dnK \times Kd}$ is defined as

$\mathbf{E}(\mathbf{W}, \mathbf{H}) :=$
$\big[ \text{vec}(\mathbf{e}_{d,1} \mathbf{e}_{K,1}^\top (\mathbf{WH} - \mathbf{Y})), ..., \text{vec}(\mathbf{e}_{d,1} \mathbf{e}_{K,K}^\top (\mathbf{WH} - \mathbf{Y})), \text{vec}(\mathbf{e}_{d,2} \mathbf{e}_{K,1}^\top (\mathbf{WH} - \mathbf{Y})), ...$
$$..., \text{vec}(\mathbf{e}_{d,d} \mathbf{e}_{K,K}^\top (\mathbf{WH} - \mathbf{Y})) \big],$$

where $\mathbf{e}_{d,i}$ is the standard vector in $\mathbb{R}^d$ with 1 in its $i$th entry (similar definition stands for $\mathbf{e}_{K,k}$).

For the collapsed minimizer $(\mathbf{W}^*, \mathbf{H}^*)$, we know that $\mathbf{H}^* = \sigma_{\overline{H}} \mathbf{R} \otimes \mathbf{1}_n^\top$ and $\mathbf{W}^* = \sigma_W \mathbf{R}^\top$ for some (partial) orthonormal matrix $\mathbf{R} \in \mathbb{R}^{d \times K}$ (i.e., $\mathbf{R}^\top \mathbf{R} = \mathbf{I}_K$).

Therefore, we have that $\mathbf{W}^* \mathbf{H}^* - \mathbf{Y} = -c\mathbf{I}_K \otimes \mathbf{1}_n^\top \otimes \mathbf{I}_d$, and that $\mathbf{H}^* \otimes \mathbf{W}^* = (1-c)\mathbf{R} \otimes \mathbf{1}_n^\top \otimes \mathbf{R}^\top$. Observe that the alignment of the former expression with the latter (where the locations of the dimensions $d$ and $K$ are swapped) is done using the matrices $\{\mathbf{e}_{d,i} \mathbf{e}_{K,k}^\top\}$. Indeed, we can write $\mathbf{E}^{*\top} = \mathbf{K}_{d,K}(-c\mathbf{I}_K \otimes \mathbf{1}_n^\top \otimes \mathbf{I}_d)$, where $\mathbf{K}_{d,K} \in \mathbb{R}^{Kd \times dK}$ is the permutation matrix that satisfies

$$\mathbf{K}_{d,K}^\top (\mathbf{X}_1 \otimes \mathbf{X}_2) \mathbf{K}_{d,K} = \mathbf{X}_2 \otimes \mathbf{X}_1$$

for any $\mathbf{X}_1 \in \mathbb{R}^{d \times d}$ and $\mathbf{X}_2 \in \mathbb{R}^{K \times K}$. Such a matrix $\mathbf{K}_{d,k}$ is also known as *commutation matrix* in the matrix theory literature. Another useful property of the commutation matrix that we will frequently use is that

$$\mathbf{K}_{d,K}(\mathbf{x} \otimes \mathbf{Y}) = \mathbf{Y} \otimes \mathbf{x} \tag{12}$$

for any $\mathbf{x} \in \mathbb{R}^{K \times 1}$ and $\mathbf{Y} \in \mathbb{R}^{d \times m}$.

Let us extract the $k, \tilde{k}$-th block $\mathbf{Z}_{k,\tilde{k}}^* \in \mathbb{R}^{dn \times dn}$ of $\mathbf{Z}^*$. First, observe that

$$\mathbf{Z}^* = (\mathbf{E}^{*\top} + \mathbf{H}^* \otimes \mathbf{W}^*)^\top (\mathbf{H}^* \mathbf{H}^{*\top} \otimes \mathbf{I}_K + n\lambda_W \mathbf{I}_{dK})^{-1} (\mathbf{E}^{*\top} + \mathbf{H}^* \otimes \mathbf{W}^*)$$

$$= \frac{1}{n} \mathbf{B}^\top (\mathbf{A} \otimes \mathbf{I}_K) \mathbf{B},$$

where
$$\mathbf{A} = (\sigma_{\overline{H}}^2 \mathbf{R}\mathbf{R}^\top + \lambda_W \mathbf{I}_d)^{-1}$$
$$\mathbf{B} = -c\mathbf{K}_{d,K}(\mathbf{I}_K \otimes \mathbf{1}_n^\top \otimes \mathbf{I}_d) + (1-c)(\mathbf{R} \otimes \mathbf{1}_n^\top \otimes \mathbf{R}^\top).$$

Denote by $\{\mathbf{e}_k\}$ the standard basis vectors in $\mathbb{R}^K$. To extract the $k, \tilde{k}$-th block of $\mathbf{Z}^*$, we compute

$$\mathbf{Z}_{k,\tilde{k}}^* = (\mathbf{e}_k \otimes \mathbf{I}_{dn})^\top \mathbf{Z}^* (\mathbf{e}_{\tilde{k}} \otimes \mathbf{I}_{dn})$$
$$= \frac{1}{n} (\mathbf{B}(\mathbf{e}_k \otimes \mathbf{I}_{dn}))^\top (\mathbf{A} \otimes \mathbf{I}_K) (\mathbf{B}(\mathbf{e}_k \otimes \mathbf{I}_{dn})),$$

with

$$\mathbf{B}(\mathbf{e}_k \otimes \mathbf{I}_{dn}) = -c\mathbf{K}_{d,K}(\mathbf{e}_k \otimes \mathbf{1}_n^\top \otimes \mathbf{I}_d) + (1-c)(\mathbf{r}_k \otimes \mathbf{1}_n^\top \otimes \mathbf{R}^\top)$$
$$= -c(\mathbf{1}_n^\top \otimes \mathbf{I}_d \otimes \mathbf{e}_k) + (1-c)(\mathbf{r}_k \mathbf{1}_n^\top \otimes \mathbf{R}^\top),$$

where in the last line, we have used property Eq. 12 to swap the Kronecker product. Then,

$$\mathbf{Z}_{k,\tilde{k}}^* = \frac{1}{n} (-c(\mathbf{1}_n^\top \otimes \mathbf{I}_d \otimes \mathbf{e}_k) + (1-c)(\mathbf{r}_k \mathbf{1}_n^\top \otimes \mathbf{R}^\top))^\top (\mathbf{A} \otimes \mathbf{I}_K)(-c(\mathbf{1}_n^\top \otimes \mathbf{I}_d \otimes \mathbf{e}_{\tilde{k}}) + (1-c)(\mathbf{r}_{\tilde{k}} \mathbf{1}_n^\top \otimes \mathbf{R}^\top))$$

$$= c^2 (\mathbf{e}_k^\top \mathbf{e}_{\tilde{k}}) \left( \frac{1}{n}(\mathbf{1}_n \mathbf{1}_n^\top) \otimes \mathbf{A} \right) + (1-c)^2 (\mathbf{r}_k^\top \mathbf{A} \mathbf{r}_{\tilde{k}}) \left( \frac{1}{n}(\mathbf{1}_n \mathbf{1}_n^\top) \otimes \mathbf{R}\mathbf{R}^\top \right)$$

$$- c(1-c) \left( \frac{1}{n}(\mathbf{1}_n \mathbf{1}_n^\top) \otimes (\mathbf{A}\mathbf{r}_{\tilde{k}} \mathbf{r}_k^\top + \mathbf{r}_{\tilde{k}} \mathbf{r}_k^\top \mathbf{A}) \right).$$

Let us write $\mathbf{R} = [\mathbf{r}_1 \mathbf{r}_2 ... \mathbf{r}_K] \in \mathbb{R}^{d \times K}$ and let $\mathbf{r}_{K+1}, ..., \mathbf{r}_d$ be the orthonormal vectors such that $\{\mathbf{r}_i\}_{i=1}^d$ forms an orthonormal basis. We know that

$$\mathbf{A} = (\sigma_{\overline{H}}^2 \mathbf{R}\mathbf{R}^\top + \lambda_W \mathbf{I}_d)^{-1} = \sum_{i=1}^K \frac{1}{\sigma_{\overline{H}}^2 + \lambda_W} \mathbf{r}_i \mathbf{r}_i^\top + \sum_{j=K+1}^d \frac{1}{\lambda_W} \mathbf{r}_j \mathbf{r}_j^\top.$$

Therefore,

$$\mathbf{r}_k^\top \mathbf{A} \mathbf{r}_{\tilde{k}} = \frac{\delta_{k\tilde{k}}}{\sigma_{\overline{H}}^2 + \lambda_W}$$

$$\mathbf{A}\mathbf{r}_{\tilde{k}} \mathbf{r}_k^\top = \mathbf{r}_{\tilde{k}} \mathbf{r}_k^\top \mathbf{A} = \frac{1}{\sigma_{\overline{H}}^2 + \lambda_W} \mathbf{r}_{\tilde{k}} \mathbf{r}_k^\top.$$

We can thus conclude that

$$\mathbf{Z}_{k,\tilde{k}}^* = \frac{1}{n}(\mathbf{1}_n \mathbf{1}_n^\top) \otimes \left( \delta_{k\tilde{k}} c^2 \mathbf{A} + \frac{\delta_{k\tilde{k}}(1-c)^2}{\sigma_{\overline{H}}^2 + \lambda_W} \mathbf{R}\mathbf{R}^\top - \frac{2c(1-c)}{\sigma_{\overline{H}}^2 + \lambda_W} \mathbf{r}_{\tilde{k}} \mathbf{r}_k^\top \right). \tag{13}$$

When $k \neq \tilde{k}$, the off-diagonal block of $\mathbf{Z}^*$ is given by

$$\mathbf{Z}_{k,\tilde{k}}^* = \frac{1}{n}(\mathbf{1}_n \mathbf{1}_n^\top) \otimes \left( -\frac{2c(1-c)}{\sigma_{\overline{H}}^2 + \lambda_W} \mathbf{r}_{\tilde{k}} \mathbf{r}_k^\top \right),$$

which is a rank-1 matrix. Since other matrices in $\mathbf{F}$ do not contribute to the inter-class block, we know that $\mathbf{F}_{k,\tilde{k}} = \frac{1}{\beta}\mathbf{Z}_{k,\tilde{k}}^*$. It is well-known that the eigenvalues of Kronecker product of two matrices are given by the products of their eigenvalues. We know that $\frac{1}{n}(\mathbf{1}_n \mathbf{1}_n^\top)$ has exactly one non-zero eigenvalue, which equals to $1$. This implies that

$$\sigma_{max}(\mathbf{F}_{k,\tilde{k}}) = \frac{2c(1-c)}{\beta(\sigma_{\overline{H}}^2 + \lambda_W)} = \frac{2\lambda_H(1 - \sqrt{\lambda_H \lambda_W})}{\beta}.$$

Next, let us compute the intra-class block. Setting $k = \tilde{k}$ in equation Eq. 13, we get

$$\mathbf{Z}_{k,k}^* = \frac{1}{n}(\mathbf{1}_n \mathbf{1}_n^\top) \otimes \left( \sum_{i=1}^d \mu_i \mathbf{r}_i \mathbf{r}_i^\top \right)$$

$$= \sum_{i=1}^d \mu_i (\frac{1}{n}\mathbf{1}_n \mathbf{1}_n^\top) \otimes (\mathbf{r}_i \mathbf{r}_i^\top)$$

where

$$\mu_k = \frac{c^2}{\sigma_H^2 + \lambda_W} + \frac{(1-c)^2}{\sigma_H^2 + \lambda_W} - \frac{2c(1-c)}{\sigma_H^2 + \lambda_W} = (2c-1)^2 \sqrt{\frac{\lambda_H}{\lambda_W}},$$

$$\mu_i = \frac{c^2}{\sigma_H^2 + \lambda_W} + \frac{(1-c)^2}{\sigma_H^2 + \lambda_W} = (c^2 + (1-c)^2)\sqrt{\frac{\lambda_H}{\lambda_W}}, \quad \text{for } 1 \le i \le K \text{ and } i \ne k,$$

$$\mu_j = \frac{c^2}{\lambda_W} = \lambda_H, \quad \text{for } K < j \le d$$

The intra-class block is therefore given by

$$\mathbf{F}_{k,k} = (1 - \frac{\lambda_H}{\beta})\mathbf{I}_{nd} - \frac{\sigma_W^2}{\beta}\mathbf{I}_n \otimes (\mathbf{R}\mathbf{R}^\top) + \frac{1}{\beta}\sum_{i=1}^d \mu_i(\frac{1}{n}\mathbf{1}_n\mathbf{1}_n^\top) \otimes (\mathbf{r}_i\mathbf{r}_i^\top)$$

$$= (1 - \frac{\lambda_H}{\beta})\sum_{i=1}^d \mathbf{I}_n \otimes (\mathbf{r}_i\mathbf{r}_i^\top) - \frac{\sigma_W^2}{\beta}\sum_{i=1}^K \mathbf{I}_n \otimes (\mathbf{r}_i\mathbf{r}_i^\top) + \frac{1}{\beta}\sum_{i=1}^d \mu_i(\frac{1}{n}\mathbf{1}_n\mathbf{1}_n^\top) \otimes (\mathbf{r}_i\mathbf{r}_i^\top)$$

$$= \sum_{i=1}^d \lambda_i\mathbf{I}_n \otimes (\mathbf{r}_i\mathbf{r}_i^\top) + \frac{1}{\beta}\sum_{i=1}^d \mu_i(\frac{1}{n}\mathbf{1}_n\mathbf{1}_n^\top) \otimes (\mathbf{r}_i\mathbf{r}_i^\top),$$

where

$$\lambda_i = 1 - \frac{\lambda_H}{\beta} - \frac{\sigma_W^2}{\beta} = 1 - \frac{1}{\beta}\sqrt{\frac{\lambda_H}{\lambda_W}}, \quad \text{for } 1 \le i \le K \tag{14}$$

$$\lambda_i = 1 - \frac{\lambda_H}{\beta}, \quad \text{for } K < i \le d \tag{15}$$

Let $\mathbf{s}_1 = \frac{1}{\sqrt{n}}\mathbf{1}_n$ and $\{\mathbf{s}_i\}_{i=1}^n$ be a set of orthonormal basis of $\mathbb{R}^n$. Then, we can further write

$$\mathbf{F}_{k,k} = \sum_{i=1}^d (\sum_{j=1}^n \lambda_i\mathbf{s}_j\mathbf{s}_j^\top) \otimes (\mathbf{r}_i\mathbf{r}_i^\top) + \frac{1}{\beta}\sum_{i=1}^d \mu_i(\mathbf{s}_1\mathbf{s}_1^\top) \otimes (\mathbf{r}_i\mathbf{r}_i^\top)$$

$$= \sum_{i=1}^d \sum_{j=1}^n \lambda_i(\mathbf{s}_j \otimes \mathbf{r}_i)(\mathbf{s}_j \otimes \mathbf{r}_i)^\top + \frac{1}{\beta}\sum_{i=1}^d \mu_i(\mathbf{s}_1 \otimes \mathbf{r}_i)(\mathbf{s}_1 \otimes \mathbf{r}_i)^\top \tag{16}$$

One can easily verify that $\{\mathbf{s}_j \otimes \mathbf{r}_i\}_{1 \le j \le n, 1 \le i \le d}$ is an orthonormal basis of $\mathbb{R}^{nd}$. So, Eq. 16 gives us the eigendecomposition of $\mathbf{F}_{k,k}$. The spectral norm of $\mathbf{F}_{k,k}$ is therefore given by

$$\sigma_{max}(\mathbf{F}_{k,k}) = \max_{1 \le i \le d}\max\{|\lambda_i|, |\lambda_i + \frac{1}{\beta}\mu_i|\}.$$

As we consider the large $\beta$ regime, the expressions in both Eq. 14 and Eq. 15 are positive. Observe that for $K < i \le d$ (associated with the over-parameterization of the model) we have that the eigenvalue associated with the eigenvector $(\mathbf{s}_1 \otimes \mathbf{r}_i)$ is given by

$$\lambda_i + \frac{1}{\beta}\mu_i = 1 - \frac{\lambda_H}{\beta} + \frac{\lambda_H}{\beta} = 1.$$

Note, though, that due to the Kronecker product with $\mathbf{s}_1 = \frac{1}{\sqrt{n}}\mathbf{1}_n$, perturbation in the direction of this eigenvector does not affect the variability in the $k$th class at all. Furthermore, generic/practical perturbations are likely to correlate with, or have their power spectrum spread over, many components of the $dn$ dimensional eigenbasis of $\mathbf{F}_{k,k}$ and not concentrate in an extremely low dimensional $d - K$ subspace (composed only of $\mathbf{s}_1 \otimes \mathbf{r}_i$ with $K < i < d$). Thus, we expect these eigenvectors to have small correlation with generic perturbations.

Showing that $\sigma_{max}(\mathbf{F}_{k,k}) = 1$ reduces now to eliminating the option of eigenvalues larger than 1 for $1 \le i \le K$. This is equivalent to having that

$$\frac{1}{\beta}\sqrt{\frac{\lambda_H}{\lambda_W}}\left(-1 + (2c-1)^2\right) < 0,$$

$$\frac{1}{\beta}\sqrt{\frac{\lambda_H}{\lambda_W}}\left(-1 + (c^2 + (1-c)^2)\right) < 0,$$

and both are ensured under our assumption $c := \sqrt{\lambda_H \lambda_W} < 1$ (the non-degenerate case of the model).

Finally, observing that Eq. 14 is smaller than Eq. 15, and that the second term in Eq. 16 does not include eigenvectors $(\mathbf{s}_j \otimes \mathbf{r}_i)$ for $j > 1$, we conclude that

$$\sigma_{min}(\mathbf{F}_{k,k}) = 1 - \frac{1}{\beta}\sqrt{\frac{\lambda_H}{\lambda_W}}.$$

### D.1 Additional Discussion on the Results of the Theorem

Theorem 4.2 has no restricting assumptions on the number of classes $K$. The only assumption, which is common in theoretical NC papers and is also what is done in practice is that $d > K$, i.e., that the dimension of the features is larger than the number of classes. This means that, regardless of the number of classes, the inter-class (off-diagonal) blocks have rank 1, while the intra-class (diagonal) blocks have full rank (recall that each block is of size $dn \times dn$).

Considering the conclusions from Theorem 4.2, which are stated in Section 4, if we sum up the maximal contribution of each of the $K-1$ inter-class blocks of a certain class, i.e., $(K-1)\sigma_{max}(\mathbf{F}_{k,\tilde{k}})$, then for guaranteeing that this sum is smaller than the minimal contribution of the intra-class block, i.e., $\sigma_{min}(\mathbf{F}_{k,k})$, we may need to assume that $\beta \gg K$. Note that this is a reasonable assumption under our large $\beta$ setting. Yet, we believe that the rank difference between the two types of blocks is a more important indicator for the dominance of the intra-class blocks, and this property is independent of the number of classes $K$. Specifically, since $dn > K$ (all the more so, in practice we even have $n \gg K$), then for generic perturbations (that uniformly span the entire $dnK$ dimensional space) the rank-1 inter-class blocks nullify much of the perturbation contrary to the intra-class block (which has full rank). This strengthen our conclusion that the deviation from collapse of each class of the minimizer $\mathbf{H}$ is dominated by the deviation from collapse of the same class in $\mathbf{H}_0$ rather than by the deviations of other classes. One thing that should be reminded here is that we analyse the "near-NC" regime, so we assume that the system is already not far from exact NC. Reaching this point in general might become harder when the number of classes grows.

Another point that can be raised regarding the results of Theorem 4.2, is that we do not analyze the full matrix $\mathbf{F}$ but rather its blocks. In fact, we believe that our analysis, which includes complete spectral analysis for each block separately, is more informative, as it clearly distinguishes between properties of intra- and inter-class blocks and provides insights on the roles of the regularization hyperparameters that are aligned with practical DNN training. In contrast, in the large $\beta$ regime we have that $\mathbf{F}$ is full rank, which masks the rank-1 property of the inter-class (off-diagonal) blocks. Nevertheless, analyzing the relationship of the full $\mathbf{F}$ and its blocks is an interesting a direction for future research.

# E ADDITIONAL EXPERIMENTS AND EXPERIMENTAL DETAILS

## E.1 EXPERIMENTAL DETAILS FOR THE LAYER-WISE EXPERIMENT

In this section, we provide the experimental details for the layer-wise training experiment that is presented in Figure 2 in the main body of the paper.

We train an MLP with 10 hidden layers on CIFAR-10 dataset, where each sample is flattened to a 3072x1 vector. Each hidden layer includes 3072 fully connected neurons with default PyTorch initialization of the weights, batchnorm, and ReLU nonlinearity. We start with one hidden layer and train the MLP with 3 epochs of Adam with mini-batch size of 256, learning rate of 1e-4, and CE loss. Then, we compute NC1 metrics for the deepest features. At this point, the first "outer iteration" of the procedure is finished. We fix the parameters in the existing hidden layers, insert a new hidden layer before the final classification layer, and repeat the procedure. Namely, at each outer iteration of the procedure we optimize only the deepest hidden layer, which has just been inserted with default PyTorch initialization of the weights, and the final classification layer, which is "initialized" with its weights from the previous outer iteration.

Let us provide more details that has led to the implementation decisions that are stated above. We have found that layer-wise training of DNNs (on a practical dataset, e.g., CIFAR-10 that we use here) is significantly harder than end-to-end training in terms of reaching a small training loss value. (Presumably, this is the reason that DNNs are typically trained in an end-to-end fashion). Careful configuration of the training procedure was required for reaching considerable low loss (though, still not zero training error) and low NC1 metrics as presented in Figure 2. From our efforts in layer-wise training the 10-layer MLP we observed the following: Adam optimizer worked better than SGD (which is harder to tune); Layer-wise minimization with CE loss (rather than MSE loss) has led to lower NC1 metrics; Using no more than 3 epochs per "outer iteration" allowed reaching lower values for the loss and the NC1 metrics at the deeper layers. Regarding the latter (i.e., more epochs per outer iteration lead to worse optimization results), when there are only one or two hidden layers then the decrease in the loss and the decrease in the NC1 metrics are larger when more epochs are being used. However, when we add in that case more hidden layers, the optimization appears to get stuck at some local minima with higher loss and NC1 metrics compared to what we get with only 3 epochs per outer iteration. As far as we understand, this behavior follows from the (extreme) nonconvexity of the problem.

## E.2 MORE EXPERIMENTS ON THE EFFECT OF THE REGULARIZATION HYPERPARAMETERS

In this section, we present more experiments that examine how modifying the regularization hyperparameters affects the NC behavior of a practical DNN – ResNet18 (He et al., 2016a) – compared to a baseline setting. Specifically, as a baseline hyperparameter setting, we consider one that is used in previous works (Papyan et al., 2020; Zhu et al., 2021): default PyTorch initialization of the weights, SGD optimizer with mini-batch size of 256, learning rate of 0.05 that is divided by 10 every 40 epochs, momentum of 0.9, and weight decay ($L_2$ regularization) of 5e-4 for all the network's parameters.

The first set of experiments is similar to the experiments in Section 5. These experiments support the insight gained in Section 4 that $\lambda_H$ (the regularization of the feature mapping) plays a bigger role than $\lambda_W$ (the regularization of the classification layer) does in approaching NC. We compare the NC1 and NC2 metrics (defined in Section 5) of the baseline setting and the following modified settings: 1) doubling the weight decay only for the last (FC) layer; 2) doubling the weight decay only for feature mapping (conv) layers; 3) zeroing the weight decay for the last layer; and 4) zeroing the weight decay for feature mapping layers.

In Figure 4 we consider the MNIST dataset with 3K training samples per class. Figure 4a presents the NC1 and NC2 metrics of the deepest features for MSE loss and no bias in the FC layer. Figures 4b and 4c present the NC1 and NC2 metrics of the deepest and intermediate (output of 3 out of the 4 ResBlock) features, respectively, when for CE loss with bias in the FC layer. In all the settings, we reach zero training error at the 40 epoch approximately. In Figure 5 we repeat the experiments with 5K training samples per class. Furthermore, repeating the experiments with 3 different random seeds for initializing the DNN's parameters yields similar curves that demonstrate the same trends. In Table 1 we report the mean and the standard deviation (SD) for the NC metrics computed for the

Table 1: The effect of modifying the weight decay (WD) on NC metrics for ResNet18 trained on CIFAR-10 and MNIST datasets – mean and SD are computed for 3 random seeds. Observe that modifying the WD in the feature mapping increases the deviation from the baseline more than modifying the WD of the last layer.

|  | CIFAR-10, MSE loss | | CIFAR-10, CE loss | | MNIST, MSE loss | | MNIST, CE loss | |
|---|---|---|---|---|---|---|---|---|
|  | NC1 | NC2 | NC1 | NC2 | NC1 | NC2 | NC1 | NC2 |
| Baseline | $0.0061 \pm 4\text{e-}4$ | $0.111 \pm 1\text{e-}2$ | $0.062 \pm 5\text{e-}3$ | $0.173 \pm 1\text{e-}2$ | $8\text{e-}4 \pm 5\text{e-}5$ | $0.072 \pm 1\text{e-}2$ | $0.004 \pm 3\text{e-}4$ | $0.115 \pm 5\text{e-}3$ |
| WDx2 for W | $0.0055 \pm 3\text{e-}4$ | $0.101 \pm 8\text{e-}3$ | $0.040 \pm 2\text{e-}3$ | $0.161 \pm 7\text{e-}3$ | $5\text{e-}4 \pm 5\text{e-}5$ | $0.055 \pm 1\text{e-}2$ | $0.003 \pm 1\text{e-}4$ | $0.102 \pm 3\text{e-}3$ |
| WDx2 for H | $0.0022 \pm 8\text{e-}5$ | $0.070 \pm 6\text{e-}3$ | $0.024 \pm 4\text{e-}3$ | $0.131 \pm 9\text{e-}3$ | $4\text{e-}4 \pm 2\text{e-}5$ | $0.048 \pm 5\text{e-}3$ | $0.002 \pm 7\text{e-}5$ | $0.101 \pm 3\text{e-}3$ |
| WD=0 for W | $0.0048 \pm 2\text{e-}4$ | $0.101 \pm 6\text{e-}3$ | $0.104 \pm 8\text{e-}3$ | $0.195 \pm 9\text{e-}3$ | $1.7\text{e-}3 \pm 1\text{e-}4$ | $0.108 \pm 2\text{e-}2$ | $0.009 \pm 4\text{e-}4$ | $0.147 \pm 5\text{e-}3$ |
| WD=0 for H | $0.0280 \pm 3\text{e-}3$ | $0.226 \pm 6\text{e-}3$ | $0.174 \pm 7\text{e-}3$ | $0.331 \pm 1\text{e-}2$ | $41\text{e-}3 \pm 2\text{e-}3$ | $0.303 \pm 2\text{e-}2$ | $0.031 \pm 4\text{e-}4$ | $0.198 \pm 8\text{e-}3$ |

deepest features at the 100 epoch (which is already after the NC metrics reach plateaus) for both the CIFAR-10 and the MNIST datasets.

Similar to previous works, from comparing Figures 4b and 4c (as well as Figures 5b and 5c) we see that the NC distance metrics are larger in the intermediate features, which correlates with the results for our model in Section 3. Examining all the settings of Figures 4 and 5, as well as Table 1, the experiments show the important role of the regularization of the feature mapping layers in approaching NC. Namely, modifying the regularization of the feature mapping layers leads to curves with larger deviations from the baseline compared to modifying the last layer's regularization. This is aligned with the theory established in Section 4 that links increasing $\lambda_H$ to reducing the dominant component of the distance from collapse of a class, which is the deviation from collapse of its own features in preceding layers.

The second set of experiments shows the role of $\lambda_W$ in mitigating the interferences between the features of different classes (such interferences can hinder approaching NC). To visualize such behavior we use a "per-class NC1" metric, defined as

$$NC_1^{(k)} := \frac{1}{K}\text{Tr}\left(\frac{1}{n}\sum_{i=1}^{n}(\mathbf{h}_{k,i} - \overline{\mathbf{h}}_k)(\mathbf{h}_{k,i} - \overline{\mathbf{h}}_k)^\top \mathbf{\Sigma}_B^\dagger\right).$$

Note that the NC1 metric, which is defined in Section 5, can be written as

$$NC_1 = \frac{1}{K}\frac{1}{K}\sum_{k=1}^{K}\text{Tr}\left(\frac{1}{n}\sum_{i=1}^{n}(\mathbf{h}_{k,i} - \overline{\mathbf{h}}_k)(\mathbf{h}_{k,i} - \overline{\mathbf{h}}_k)^\top \mathbf{\Sigma}_B^\dagger\right) = \frac{1}{K}\sum_{k=1}^{K}NC_1^{(k)}.$$

We also use the following metric to measure the alignment of the mean features and the last layer's weights

$$NC_3 := \left\|\frac{\mathbf{W}(\overline{\mathbf{H}} - \overline{\mathbf{h}}_G \mathbf{1}_K^\top)}{\|\mathbf{W}(\overline{\mathbf{H}} - \overline{\mathbf{h}}_G \mathbf{1}_K^\top)\|_F} - \frac{1}{\sqrt{K-1}}(\mathbf{I}_K - \frac{1}{K}\mathbf{1}_K \mathbf{1}_K^\top)\right\|_F,$$

where the simplex ETF is normalized to unit Frobenius norm.

In Figure 6a we present the NC metrics of the deepest features of the baseline training scheme on the MNIST dataset with 3K samples per class. The other lines in Figure 6 show the NC metrics for a modified training set, where the samples of classes (digits) 4 and 9 are degraded by a uniform blur (blur kernel of size $9 \times 9$) that hardens the distinction between them. Each line corresponds to a different value of weight decay for the last layer's parameters. Yet, in all of the settings we reached zero training error at the 40 epoch approximately. The empirical results show that large $\lambda_W$ facilitates reaching reduced NC metrics (closeness to NC structure) by reducing the effect ("interference") of the features of the degraded samples on the features of the other classes. This is aligned with the theory that is established for our model in Section 4.

## F  ADDITIONAL MOTIVATION FOR THE MODEL IN EQ. 2

In the model that we consider in Eq. 2, we interpret $\mathbf{H}$ as the deepest features of a DNN and $\mathbf{H}_0$ as shallower features of the DNN. In particular, in the large $\beta$ regime that we theoretically analyze in the paper, we interpret $\mathbf{H}_0$ as the penultimate features (one layer before $\mathbf{H}$). Even though the relation between $\mathbf{H}$ and $\mathbf{H}_0$ in our model differs from their explicit relation in many practical DNNs, there exist networks where it is very reasonable to assume that the deepest features and the penultimate features are close to each other.

For example, consider the ResNet architecture from (He et al., 2016b), where (under our interpretation of $\mathbf{H}$ and $\mathbf{H}_0$) the deepest features obey $\mathbf{H} = \mathbf{H}_0 + \mathbf{r}(\mathbf{H}_0)$, where $\mathbf{r}(\cdot)$ denotes a residual block. The residual term can potentially be very small if $\mathbf{H}_0$ already separates the classes (e.g., it has a "near NC" structure). In fact, in the popular neural ODE framework (Chen et al., 2018), which is understood as the infinite depth limit of these ResNets, we inherently have that $\mathbf{H} \approx \mathbf{H}_0$. Another example where the concept $\mathbf{H} \approx \mathbf{H}_0$ inherently holds is deep equilibrium models (DEQ) (Bai et al., 2019). These practical DNN frameworks provide the rationality for analyzing our model. Furthermore, our theoretical results, such as depthwise decrease in the within-class variability, are aligned also with the empirical behavior of DNN architectures beyond the aforementioned examples (e.g., plain MLP).

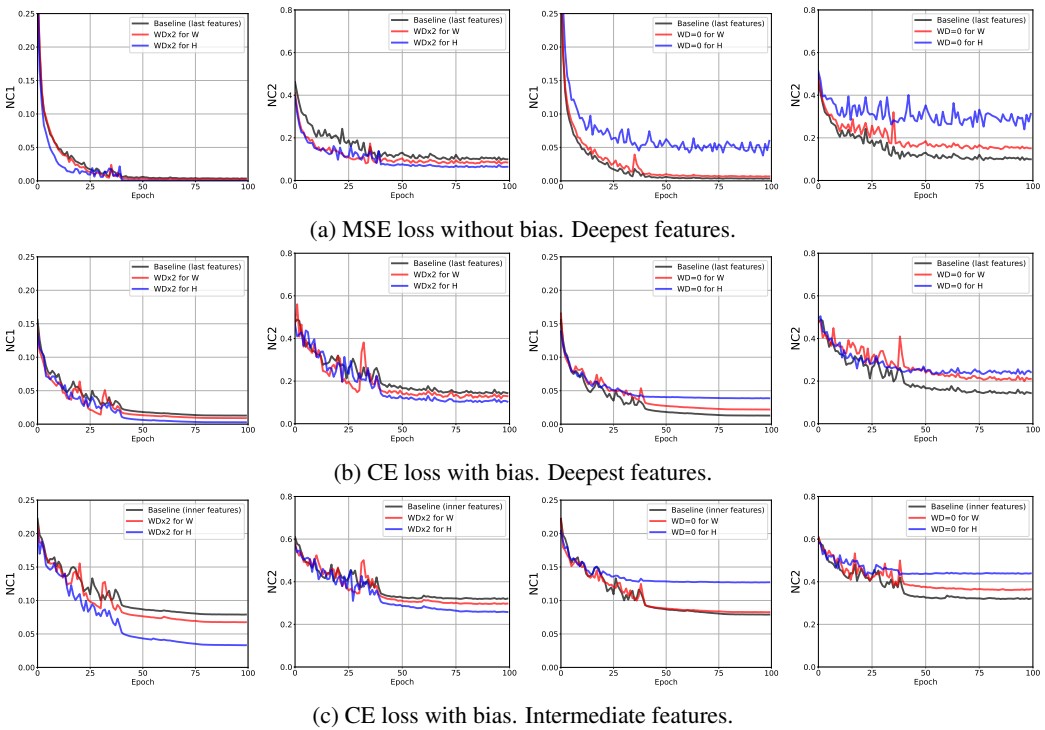

(a) MSE loss without bias. Deepest features.

(b) CE loss with bias. Deepest features.

(c) CE loss with bias. Intermediate features.

Figure 4: The effect of modifying the weight decay (WD) on NC metrics for ResNet18 trained on MNIST with 3K samples per class. Observe that modifying the WD in the feature mapping increases the deviation from the baseline more than modifying the WD of the last layer.

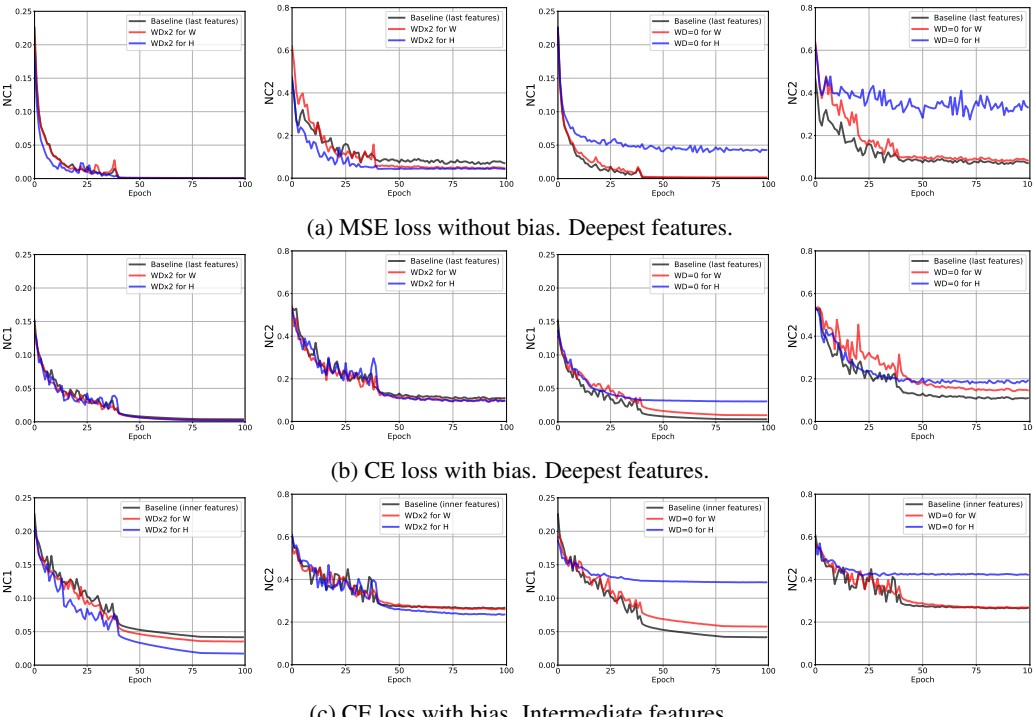

(a) MSE loss without bias. Deepest features.

(b) CE loss with bias. Deepest features.

(c) CE loss with bias. Intermediate features.

Figure 5: The effect of modifying the weight decay (WD) on NC metrics for ResNet18 trained on MNIST with 5K samples per class. Observe that modifying the WD in the feature mapping increases the deviation from the baseline more than modifying the WD of the last layer.

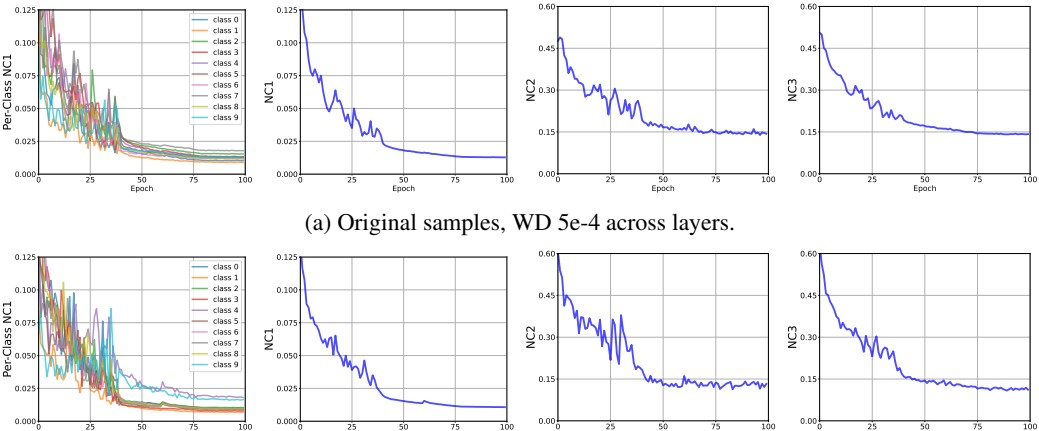

(a) Original samples, WD 5e-4 across layers.

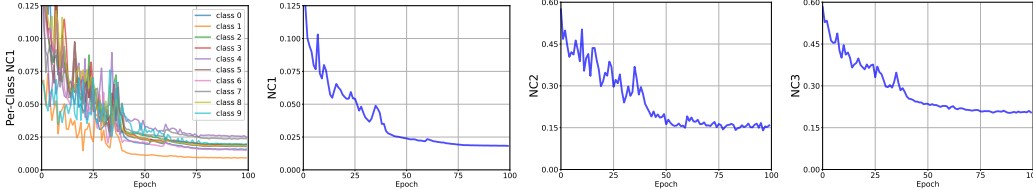

(b) Samples of classes 4 and 9 are blurred, last layer's WD remains 5e-4. The effect of the blurred classes on the NC metrics (avg. and other classes) is minor.

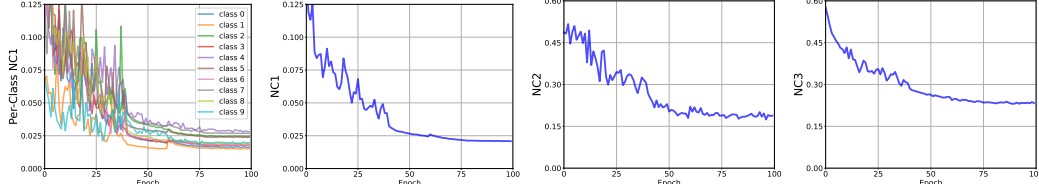

(c) Samples of classes 4 and 9 are blurred, last layer's WD reduced to 5e-5. The blurred classes affect the "per-class NC1" of other classes and the NC metrics increase.

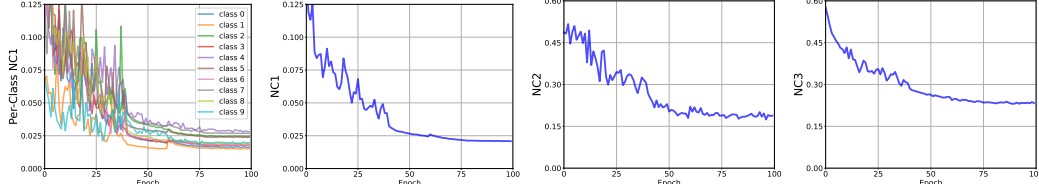

(d) Samples of classes 4 and 9 are blurred, last layer has no WD. The blurred classes further interfere with other classes and the NC metrics further increase.

Figure 6: The effect of modifying the weight decay (WD) of the last layer's weights on NC metrics for ResNet18 trained on MNIST with 3K samples per class where *samples from classes 4 and 9 are blurred*. Observe that small WD in the last layer increases the effect of the "pre-class NC1" curves of the blurred classes on the other classes, and increases also the other NC metrics.

