# OpenReview forum: "Perturbation Analysis of Neural Collapse"
_ICLR.cc/2023/Conference — Submitted to ICLR 2023_

### Official Review · Reviewer_8Wp5 · 2022-10-25

**Confidence:** 3
**Correctness:** 3
**Technical Novelty And Significance:** 3
**Empirical Novelty And Significance:** 2
**Recommendation:** 5

**Clarity, Quality, Novelty And Reproducibility:**

The paper is reasonably clearly written, though as I have commented previously does have a number of typos. I have not had an opportunity to review the proofs in depth, but I didn't notice anything obviously incorrect at first glance. My main concern with the paper is the relevance of the theoretical model to the real-world phenomenon it is seeking to describe. As is often the case in theoretical deep learning papers, it is difficult to determine whether the qualitative agreement of experiments with theory is due to the correctness of the theory as a model for the empirical phenomenon or due to the theory being constructed so as to describe observations of situations that are similar to the experiments being conducted.

**Strength And Weaknesses:**

Strengths

- The model studied in this paper is more realistic than previous works that have completely unconstrained features.
- The paper is clearly written.
- The modified trace term measuring neural collapse defined in Section 3 is well justified, and I appreciated the comparison against prior works to highlight the necessity of $\widetilde{NC}_1$ in Theorem 3.1
- The perturbation analysis in Section 4 provides a nice intuition for how non-collapsed features in early layers of the network can slow down convergence to NC in later layers.
- The experiments are easy to interpret and provide a sensible evaluation of the theoretical predictions.
- The visualization of the operator $F$ studied in Theorems 4.1 and 4.2 is helpful to ground the discussion around these results.

Weaknesses/Questions:
- I’m unsure how realistic the gradient flow in (4) is as a model for feature learning in neural networks.
In moving from the penalty on the distance $\|H - H_0\|$ to the gradient flow, the analysis seems to implicitly change the interpretation of $H_0$ as the output of a previous layer to the value of the current features.
- Further, this requires that the weights always be optimal with respect to the current features. This assumption seems unlikely to be true in neural networks, and it’s not clear to me how the approximation error and resulting dynamics attained by suboptimal weights would relate to the “central path” described in this paper.
- The conclusions from Theorem 4.2 seem to implicitly assume a relatively small number of classes in order for the contribution of the inter-class blocks of $\mathbf{F}$ on the off-diagonal to be negligible. I would be interested in seeing (at least in the appendix) a discussion of how the relative contribution of inter- and intra-class interactions vary as a function of the number of classes. In particular, how do the spectra of the blocks translate to the spectrum of the overall linear operator $F$? While I appreciate the empirical example in Figure 1, I would have liked to see a more detailed theoretical analysis of the relationship between the sub-blocks and the overall matrix.

Minor errata:
“Values” is misspelled twice as “vlaues”.


**Summary Of The Paper:**

This paper presents a new model for neural collapse (NC) which penalizes the degree to which the learned features diverge from some initial value. This model, it is argued, more closely aligns with the evolution of features from one layer to the next in a neural network. This modification to the unconstrained features model, coupled with the assumption that the weight vector is optimal with respect to the features, yields a gradient flow which also exhibits neural collapse. Finally, a perturbation analysis is applied to the model when the previous-layer features are not collapsed. The principal take-away from this analysis is that it is intra-class variation, rather than inter-class variation, of the previous-layer features which drives the deviation from collapse of the output features.

**Summary Of The Review:**

This paper presents an interesting perspective on neural collapse by studying the behaviour of networks that are near, but do not precisely exhibit, feature collapse. The theoretical model used in this analysis comes closer to capturing the limited ability of neural networks to arbitrarily change the feature representation, but still makes strong assumptions on e.g. the optimality of the weights $W$ over the course of training and the evolution in the value of the initial earlier layer features. The perturbation analysis results are intriguing and provide some intuition for how features may exhibit slow convergence to total collapse, but the closed-form solution obtained by these results is quite unwieldly and difficult to interpret except through analysis of sub-matrices. Overall, I think this paper makes a step towards studying neural collapse in a more realistic regime, but there remains a wide gap between the theoretical model studied here and practical settings which limits the significance of the contribution.

---

> ### Author Response · Authors · 2022-11-15
> **Authors Response (1/2)**
>
> We thank the reviewer for handling our paper. Below is our point-to-point response to the comments.
>
> - *Rationality of the gradient flow and the penalty term*
>
> As explained in the paper, we use the gradient flow that is stated in Eq. (4), which is associated with the UFM in Eq. (1) to obtain results *that are rigorously translated* in Corollary 3.2 to results on the minimizer of Eq. (2) in the large $\beta$ regime (even though the
> subscript of $H_0$ is interpreted differently in the two settings).
>
> Please notice that in the revision we also improve the motivation for the model in Eq. (2) (in Section 2 and Appendix F).
> Specifically, we refer to cases of practical DNNs (a ResNet type, neural ODE, deep equilibrium models) where the distance between between the deepest features ($H$ in Eq. (2)), and the preceding (penultimate) features ($H_0$ in Eq. (2)) may be small or even inherently small.
> These practical models further motivate analyzing our model.
> Yet, our theoretical results, such as depthwise decrease in the within-class variability, are aligned also with the empirical behavior of DNN architectures beyond the aforementioned examples (e.g., the plain MLP in Figure 2).
>
>
> - *The optimal W=W\*(H) setting*
>
> Analyzing the objective of an MSE-based UFM (e.g., the one in Eq. (1)) as a function of $H$ alone, by exploiting the fact that the minimizer w.r.t. $W$ is a function of $H$, i.e., $W=W^*(H)$, is the focus of the entire paper (Han et al., 2022 [1]) that has won the ICLR 2022 Outstanding Paper Prize.
> As we stated in our paper: The motivation for studying such an objective, where the optimization variable $W$ is replaced by the optimal $W^*(H)$, comes from the empirical observation (via extensive experimentation in (Han et al., 2022 [1])) that the gap $\lVert W^*(H)H-Y \rVert_F^2 - \lVert W H-Y \rVert_F^2$ is small (compared to each term) during the optimization process of practical DNNs.
>
> This justifies exploring this setting and highlights the significance of our gradient flow theory.
> Furthermore, note that the optimality $W=W^*(H)$ is a trivial property (and not an assumption) of the minimizer of Eq. (2) which is the main object that we analyze in our paper.

---

> > ### Author Response · Authors · 2022-11-15
> > **Authors Response (2/2)**
> >
> > - *The conclusions from Theorem 4.2*
> >
> > Regarding the effect of increasing $K$,
> > note that Theorem 4.2 has no restricting assumptions on the number of classes.
> > The only assumption, which is common in theoretical NC papers and is also what is done in practice is that $d>K$, i.e., that the dimension of the features $d$ is larger than the number of classes $K$.
> > This means that, regardless of the number of classes, the inter-class (off-diagonal) blocks are still only rank 1. Recall that each block is of size $dn \times dn$ and that the intra-class (diagonal) blocks are full rank.
> >
> > Focusing on the conclusions from Theorem 4.2, the reviewer is right that if
> > we sum up the maximal contribution of each of the $K-1$ inter-class blocks of a certain class, i.e., $(K-1) \sigma_{max}(F_{k,\tilde{k}})$, then for guaranteeing that this sum is smaller than the minimal contribution of the intra-class block, i.e., $\sigma_{min}(F_{k,k})$, we may need to assume that $\beta \gg K$, which is a reasonable under our large $\beta$ setting. We thank the reviewer for raising this point and mention it in the revision.
> > Yet, we claim that the rank difference between the two types of blocks should not be overlooked.
> > Since $dn>K$ (all the more so, in practice we even have $n \gg K$), we have that for generic perturbations (that uniformly span the entire $dnK$ dimensional space) the rank-1 inter-class blocks nullify much of the perturbation contrary to the intra-class block (which is full rank).
> > This strengthen our conclusion that the deviation from collapse of each class of the minimizer $H$ is dominated by the deviation from collapse of the same class in $H_0$ rather than by the deviations of other classes.
> > One thing that should be reminded here is that we analyze the ``near-NC" regime, so we assume that the system is already not far from exact collapse. Reaching this point in general might become harder when the number of classes grows.
> > To conclude, we thank the reviewer for this comment and add this discussion to Appendix D.1 in the revision.
> >
> > Regarding spectral analysis of the full matrix $F$ compared to spectral analysis of its blocks, we actually believe that the current situation where we have complete spectral analysis for each block separately is much more informative, as it clearly distinguishes between properties of intra- and inter-class blocks and provides insights on the roles of the regularization hyperparameters that are aligned with practical DNN training.
> > Therefore, we believe that our current theory is already a significant contribution.
> >
> > As far as we know, in general, complete spectral analysis of blocks of a matrix cannot be translated to complete spectral analysis of the full matrix.
> > We are also not sure what is the gain from such analysis for the full matrix $F$ compared to the complete per-block analysis that we have.
> > For example, clearly in the large $\beta$ regime we have that $F$ is full rank, which masks the rank-1 property of the inter-class (off-diagonal) blocks.
> > In the appendix of the revision, we include such a discussion and suggest analyzing the relationship between the full $F$ and its sub-blocks as a direction for future research.
> >
> >
> > - *Response to the summary*
> >
> > As we explained in our answers above,
> > the assumption on the optimality of $W$ (as a function of $H$) in the gradient flow analysis is reasonable and using it when analyzing DNNs is justified by extensive experimentation in (Han et al., 2022 [1]).
> > Furthermore, when analyzing the minimizer of our model (Eq. (2)) it is a property and not an assumption.
> > We also discussed the benefits of per-block spectral analysis in the ``near-NC" regime. For example, analyzing the entire matrix $F$ masks the low rank property of the off-diagonal blocks.
> > In Section 2 and Appendix F of the revision we also add more motivation for the considered model (Eq. (2)).
> >
> >
> > - *References*
> >
> > [1] XY Han, Vardan Papyan, and David L Donoho. Neural collapse under MSE loss: Proximity to and dynamics on the central path. In International Conference on Learning Representations (ICLR), 2022.

---

> > > ### Comment · Reviewer_8Wp5 · 2022-11-27
> > > **Response**
> > >
> > > Thanks to the authors for their response. I appreciate the author's clarification on the types of DNN architectures that we can expect to be well-approximated by the model, along with the discussion of the evidence provided in prior work to justify the assumed optimality of the final-layer weights. The discussion of the block structure, and of the need to balance $\beta$ and $K$, is helpful but my concern regarding the translation of the spectral analysis of the blocks of F to the behaviour of the full matrix remains. As the authors mention in their response, it is often the case that the behaviour/spectrum of a large matrix may differ significantly from that of its sub-matrices, and so I am not fully convinced that this style of analysis will fully capture the behaviour of F.

---

### Official Review · Reviewer_1Pqf · 2022-10-25

**Confidence:** 4
**Correctness:** 3
**Technical Novelty And Significance:** 2
**Empirical Novelty And Significance:** 2
**Recommendation:** 3

**Clarity, Quality, Novelty And Reproducibility:**

Clarity and quality of the paper decrease with the repetitively made approximations, and the inflated math. The theorems are quite difficult to read (for example Thm. 4.1, what is $F_{k,k}$ in Thm. 4.2) and the meaning of these theorems is also not obvious to me. As already pointed out, the novelty is also low because most of the derived intuitions/relations about solvers of the matrix factorization objective are already known. Reproducibility is ok.

**Strength And Weaknesses:**

The main weakness of this paper and the referenced paper on which the authors build (Tirer&Bruna, 2022) is that most of the results and contributions already have been studied in the field of matrix factorization. This whole field is entirely neglected and properties of the objective solvers are derived over approximations of approximations, that just as well follow directly from the theory of matrix factorization (using maybe even a more realistic explanation, see below). Considering this, the contributions of this paper seem thin.

I think that the introduced neural collapse score is intuitive and that it could be useful. I also think that there is some insight gained from the experiments confirming that sparsity in the learned transformed feature representation induces neural collapse. Although I have some doubts whether the experiments actually correspond to the theory.

# Assumptions and Approximations
There are quite a few assumptions that make the derived theory less impactful. First of all, analyzing the matrix factorization model as a substitute for neural network optimization is very doubtful. After all, the targets (class labels) are not directly approximated by a linear function, but by the softmax of a linear function. Then, cross-entropy is typically used instead of MSE. In addition, the authors make the assumption that each class has exactly $n$ samples. To make the $l_2$-norm regularized matrix factorization model of Tirer & Bruna more "realistic", the authors introduce the regularization term $\lVert H-H_0\rVert^2$. This is an approximation for the fact that $H$ can not be chosen freely, but is instead a function of the input, given by the neural network's representations learned as the output of the penultimate layer. So, this penalty term is quite far from what's happening in reality and I also don't see how the provided analysis of this objective can actually provide insight.

To prove Theorem 3.1, the gradient flow is analyzed of, again, an approxiation $\frac{dH_t}{dt} = -Kn\nabla\mathcal{L}(H_t)$. In fact, we only have   $\frac{H_t-H_0}{t} = -Kn\nabla\mathcal{L}(H_t)$.

# The Results from a Matrix Factorization Perspective
I think that the most interesting result of this paper is the analysis of the within and between class scatter in dependence of the sparsity penalization of $H$. The theorems in this paper don't show exact properties but rely on multiple approximations. The approximate result can be also directly derived from the theory of matrix factorization. Considering that we actually analyze now $\lVert Y-WH\rVert$, then  after the penultimate layer, most architectures use a ReLU activation. Hence, $H$ is nonnegative. If we restrict $H$ to be a one-hot encoded matrix, that is $H\in\\{0,1\\}^{d\times Kn}$ and $\lVert H_{\cdot i}\rVert=1$, then the objective $\min_{W,H}\lVert Y-WH\rVert$ is _equivalent_ to $k$-means on the matrix $Y$, where $W$ indicates the centroids. Hence, trivially, this objective would minimize the within-class scatter and maximize the between-class scatter. A similar result can however be achieved if we introduce a sparsity constraint on $H$. the nonnegative, sparse matrix $H$ indicates in this case a fuzzy clustering, where the nonzero elements in $H_{k j}$ indicate a degree with which the  sample $x_j$ belongs to class $k$. The higher the sparsity regularization weight, the fewer classes are selected for a sample to belong to, and the closer we get to the $k$-means clustering.

# Experiments
The experiment results are depicted in Figure 3. A network is trained layer-wise on Cifar 10. The sparsity constraint on $H$ is simulated by a weight decay on all weights between layers, except for the last one. This is however not the same as sparsity in the learned representations given by $H$, which is the output of the penultimate layer. The experiments indicate that a higher weight decay on "H" increases the neural collapse metric on Cifar-10, which is an interesting result, but it only loosely connects to the theory.

## References
Udell, Madeleine et al. “Generalized Low Rank Models.” Found. Trends Mach. Learn. 9 (2016): 1-118. (see in particular "2.2 quadratically regularized PCA" and "3.2 Examples" )

**Summary Of The Paper:**

The authors analyze properties of a sparsity regularized matrix factorization where one of the factor matrices $H$ is further regularized to be close to a given matrix $H_0$:
\begin{align} \min_{W,H}f(W,H) = \lVert Y-WH \rVert^2 +\lambda_W\lVert W\rVert^2 +\lambda_H \lVert H\rVert^2 + \beta \lVert H-H_0\rVert\end{align}
The properties of the solvers of the above objective shall give insight into properties of neural collapse, describing the phenomenon that neural networks tend to resemble nearest centroid-classifiers after training, where the class centroid is computed in the penultimate layer. The matrix $Y\in\\{ 0,1 \\}^{K\times Kn}$ indicates here the one-hot encoded targets for $K$ classes and $n$ samples per class. $H\in\mathbb{R}^{d\times Kn}$ is in this case related to the $d$-dimensional output of the penultimate layer and the matrix $H_0$ indicates architectural restrictions on what kind of representations can be learned. The matrix $W$ is in return related to the weight matrix connecting the penultimate layer with the last layer.

The main result (Thm 1) indicates (approximately) that when $\lambda_W>0$ the within-class scatter decreases and the between-class scatter increases with $\beta\rightarrow 0$.
Experiments evaluate how the weight decay on specific layers influences the neural collapse behavior on Cifar 10.

**Summary Of The Review:**

The authors provide support to the hypothesis that sparsity in the learned representations of neural networks, given by the output of the penultimate layer, induces/strengthen the behavior of neural collapse. The provided theory is only loosely connected to the reality of neural network training and is also not novel in the light of matrix factorization theory. Experiments are also just loosely simulating the sparsity in learned representations, which makes the results overall not very convincing.

# After Rebuttal Thoughts
I did not find the rebuttal very convincing. The theoretical analysis is still weak. Even when the network is trained with MSE loss, the objective function would not be the considered objective, but softmax would be applied on the matrix product. Generally, the nonlinear parts of the objective and the learned function, represented by $H$ are neglected. The analysis of the considered objective does still not deliver more knowledge than what is known from matrix factorization theory. The use of gradient flow is not a contribution in itself, it doesn't really fit here (see my criticism on approximations). The addition of the penalty term is emphasized as a contribution, but the anlysis focuses on the case where the weight of the penalty term goes to zero. Hence, I keep my score and I strongly encourage to have a look into the connection of NMF with clustering methods to improve further work.

---

> ### Author Response · Authors · 2022-11-15
> **Authors Response (1/n)**
>
> We thank the reviewer for handling our paper, though we disagree with his main criticism. Below is our point-to-point response to the comments.
>
> - *Claim on "main weakness"*
>
> We disagree with the reviewer's claims that "most of the results and contributions already have been studied in the field of matrix factorization" or that our results "follow directly from the theory of matrix factorization".
> **As we explain below (and in the following answers), these claims are wrong.**
>
> The general form of the model in Eq. (1) may be similar to models studied in matrix factorization literature, but since the latter models typically assume more rows than columns in the left matrix $W$ (e.g., low-rank factorization) and do not focus on the specific one-hot matrix structure of $Y$ they do not provide detailed characterization of the minimizers as the theorem taken from (Tirer & Bruna, 2022).
> And anyway --- in this paper we have major contributions regarding *a different model*, i.e., the model in Eq. (2), which has an additional penalty term that has a critical effect on the problem.
>
> The contributions of our paper are novel and significantly different than results in exiting literature.
> We have not encountered in the literature any result that is similar to our gradient flow theory, translation of this theory to our model in Eq. (2), and spectral analysis of the effects of perturbations in ``near-NC" regime of the model in Eq. (2).
> We mention matrix factorization in our revision when we present Eq. (1), but we do not see any way to extract the results of our paper from anything currently appearing in the literature.
>
> The claim on the similarity of our work to literature on matrix factorization, which is further discussed by the reviewer below (and addressed by us below), considers a different model without the penalty term that relates $H$ to $H_0$,
> with different assumptions on $H$, and with intuitions (for his different model) that are no close to our formal analysis of the near-NC case which is obtained by Taylor-based approximation and detailed spectral decomposition.
> Therefore, this criticism is wrong.
>
>
> - *Assumptions and Approximations*
>
> (Hui & Belkin, 2021 [1]) have shown that training DNN classifiers with MSE loss (as considered in our analysis) is a powerful strategy that yields results similar to (and sometimes better than) training with cross-entropy (CE) loss.
> Their paper, and its follow ups, justify theoretical papers of NC that consider MSE loss, such as (Han et al., 2022 [2]), (Tirer & Bruna, 2022 [3]), and (Zhou et al., 2022 [4]).
> Furthermore, note that we empirically show that our theoretical insights (layer-wise decrease in NC1 metric, effects of regularization hyperparameters) are aligned with the behavior of DNNs that are trained with CE loss.
> Nevertheless, following this comment, we highlight this motivation for considering the MSE loss in the revision.
>
> The assumption that different classes have the same number of samples in also very reasonable for theoretical works. It also holds in some practical datasets (e.g., CIFAR-10 and CIFAR-100), and even for a dataset like MNIST, where this assumption does not exactly hold, our experiments show agreement between the theoretical findings and the empirical behavior of DNNs.
>
> Regarding the comment on the motivation for the term $\lVert H-H_0 \rVert_F^2$, indeed, in the model that we consider (Eq. (2)) the relation between $H$, which we interpret as the deepest features, and $H_0$, which we interpret as the preceding (penultimate) features, differs from their explicit relation in many practical DNNs.
> Yet, there exist also networks where it is very reasonable to assume that the deepest features and the penultimate features are close to each other. Specifically, consider the ResNet architecture of (He et al., 2016 [5]) where (under our interpretation of $H$ and $H_0$) the deepest features obey $H = H_0 + r(H_{0})$, where $r(\cdot)$ denotes a residual block. The residual term can potentially be very small if $H_0$ already separates the classes (e.g., it has a ``near NC" structure).
> In fact, in the popular neural ODE framework (Chen et al., 2018 [6]), which is understood as the infinite depth limit of these ResNets, we inherently have that $H \approx H_{0}$.
> Another example where the concept $H \approx H_{0}$ inherently holds is deep equilibrium models (DEQ) (Bai et al., 2019 [7]).
> These practical models provide the rationality for analyzing our model.
> Furthermore, our theoretical results, such as depthwise decrease in the within-class variability, are aligned also with the empirical behavior of DNN architectures beyond the aforementioned examples (e.g., the plain MLP in Figure 2).
> This discussion, which motivates our model, is added to Section 2 and Appendix F of the revision.

---

> > ### Author Response · Authors · 2022-11-15
> > **Authors Response (2/n)**
> >
> > - *Comment on the usage of Theorem 3.1*
> >
> > Corollary 3.2 rigorously translates results that are obtained for the gradient flow in Theorem 3.1 to results on the minimizer of Eq. (2) in the large $\beta$ regime.
> > Examining more general regimes of $\beta=1/t$ is an interesting direction for future research.
> >
> > Note again that Theorem 3.1 is a very significant contribution on it own. As stated in the paper: it explores the gradient flow of a UFM, which has been motivated and empirically justified in the ICLR award-winning paper (Han et al., 2022 [2]), yet compared to the analysis in (Han et al., 2022 [2]) it has only minimal assumptions and no engineered modification of the gradient flow (specifically, it avoids engineered renormalization and projection of the gradient).
> >
> >
> > - *Matrix Factorization Perspective*
> >
> > First of all, it is not clear what the reviewer refers to in "sparsity".
> > Typically, sparsity refers to having a small number of entries that are different than zero. However, in the model that we consider the minimizer w.r.t. the features matrix $H$ and the predefined features $H_0$ are not sparse in that sense (even in the case of exact NC, the matrices are typically non-sparse in that sense).
> > Furthermore, the regularization of $H$ through $\lVert H \rVert_F^2$ and its penalty through $\lVert H-H_0 \rVert_F^2$ are not terms that promote sparsity in the sense of having many zeros.
> > Similarly to the vector case, where $\lVert h \rVert_2^2$ does not promote sparsity, contrary to $\lVert h \rVert_1$.
> > Restricting $H$ to have only one-hot columns is completely different than the considered model.
> > The reviewer also ignores the penalty term that relates $H$ to $H_0$ which is a critical part of our model.
> > Later in his comment, when the reviewer tries to drop his assumption that $H$ has only one-hot columns he presents
> > "intuitions" (still without considering any term with $H_0$) that are *no close to* our formal analysis of the near-NC case which is obtained by Taylor-based approximation and detailed spectral decomposition.
> > Finally, even if the reviewer tries to make his claims specifically against our gradient flow theory, i.e., Theorem 3.1 (which is only a part of our contributions and contrary to the other parts does not include the $\lVert H-H_0 \rVert_F^2$ term),
> > note that even if certain behavior seems "intuitive", this is by no means obvious either at a purely formal or at a rigorous level. All the more so, intuitive explanation cannot provide the rates of the dynamics.
> >
> > - *Experiments*
> >
> > The reviewer again seems to interpret the L2-regularization of $H$ in our model as sparsity, which is unclear.
> > Regarding the regularization of $H$, as we state in the paper:
> > "Based on the equivalence of L2-regularization with weight decay (WD) in gradient-based methods, we can make the analogy of regularizing $H$ in Eq. (2) to WD of the weights of practical DNNs in the feature mapping layers (i.e., excluding the last layer’s weights). *Importantly*, note that this analogy is empirically justified  in (Zhu et al., 2021 [8])." Specifically in Figure 6 in the arxiv version of their paper (or look at their Neurips21 supp. mat.) they show similar behavior with the two types of regularizations.
> > Thus, the empirical results are well connected with our theory in Section 4, which clearly shows that in the ``near-NC regime" large regularization of the feature mapping promotes closeness to exact NC structure.
> >
> >
> > - *Comment on "Clarity, Quality, Novelty"*
> >
> > Note that all the other reviewers generally complimented our writing and presentation.
> > Regarding Theorem 4.2, it clearly tells the reader to consider the representation of Eq. (5) that is given in Eq. (6). We have a full paragraph just above Theorem 4.2 that explains in detail the representation that is given in Eq. (6), including each of the blocks $F_{k,k}$ and $F_{k,\tilde{K}}$.
> >
> > Again, the criticism that "the novelty is also low because most of the derived intuitions/relations about solvers of the matrix factorization objective are already known" is wrong.

---

> > > ### Author Response · Authors · 2022-11-15
> > > **Authors Response (3/ n=3)**
> > >
> > > - *References*
> > >
> > > [1] Like Hui and Mikhail Belkin, "Evaluation of neural architectures trained with square loss vs cross-entropy in classification tasks." In The Ninth International Conference on Learning Representations (ICLR). 2021.
> > >
> > > [2] XY Han, Vardan Papyan, and David L Donoho. Neural collapse under MSE loss: Proximity to and dynamics on the central path. In International Conference on Learning Representations (ICLR), 2022.
> > >
> > > [3] Tom Tirer and Joan Bruna. Extended unconstrained features model for exploring deep neural collapse. In Proceedings of the 39th International Conference on Machine Learning (ICML), volume 162, pp. 21478–21505. PMLR, 2022.
> > >
> > > [4] Jinxin Zhou, Xiao Li, Tianyu Ding, Chong You, Qing Qu, and Zhihui Zhu. On the optimization landscape of neural collapse under MSE loss: Global optimality with unconstrained features. In Proceedings of the 39th International Conference on Machine Learning (ICML), volume 162, pp. 27179–27202. PMLR, 2022.
> > >
> > > [5] He, Kaiming, Xiangyu Zhang, Shaoqing Ren, and Jian Sun. "Identity mappings in deep residual networks." In European conference on computer vision, pp. 630-645. Springer, Cham, 2016.
> > >
> > > [6] Chen, Ricky TQ, Yulia Rubanova, Jesse Bettencourt, and David K. Duvenaud. "Neural ordinary differential equations." Advances in neural information processing systems 31 (2018).
> > >
> > > [7] Bai, Shaojie, J. Zico Kolter, and Vladlen Koltun. "Deep equilibrium models." Advances in Neural Information Processing Systems 32 (2019).
> > >
> > > [8] Zhihui Zhu, Tianyu Ding, Jinxin Zhou, Xiao Li, Chong You, Jeremias Sulam, and Qing Qu. A geometric analysis of neural collapse with unconstrained features. Advances in Neural Information Processing Systems (NeurIPS), 34:29820–29834, 2021.

---

### Official Review · Reviewer_aJ5E · 2022-10-26

**Confidence:** 3
**Correctness:** 3
**Technical Novelty And Significance:** 3
**Empirical Novelty And Significance:** 1
**Recommendation:** 6

**Clarity, Quality, Novelty And Reproducibility:**

The paper is in general clearly written, theoretical results are coherently stated and experimental section is explained sufficiently.

**Strength And Weaknesses:**

Strengths:
-----------------------------------------------

1. The motivating hypothesis is interesting- whether the widely used UFM is sufficient for addressing the NC1 behavior in DNNs
2. The authors propose a simple modification to the UFM to penalize deviation from a fixed feature matrix. A justification on its usefulness is provided by connecting the effect to that of finite depth neural networks not permitting the last layer features to wander freely. This can prove to be a useful model for relaxing the infinite approximation capability of neural networks that is proposed as a heuristic justification for studying the unconstrained features model.
3. The authors prove two main properties of the optimum of the proposed modified UFM under MSE loss: the monotonic reduction in NC1, effect of regularization on the deviation. These are consistent with observed behavior in DNN training.

Weaknesses:
-----------------------------------------------

1. It is not clear that the hypothesis of NC1 not being 0 is completely justified- can this be proved in a simple model, where when SGD is performed, the “final” solution does not exhibit exact collapse? Is the experimental observation strong enough to eliminate other effects such as not being deep in the terminal phase of training? For example, NC1 is Fig. 3 top row has a value less than 1e-2. For practical purposes it might be sufficient to say that the features have collapsed. Are there significant and specific consequences of the metric not being much smaller?
2. I think more extensive and careful experimentation can make a stronger case for the proposed model to be useful as a strictly better model than the simple UFM.
3. On the layer-wise experiment: this is an interesting study. However, from the details in the appendix, it seems like each layer is trained for only 3 epochs- if run longer, the layer would have likely achieved smaller NC1, taking the setting closer to near-collapse. Any specific reason for stopping at 3 epochs? Also, what is the reason for using Adam rather than SGD for this experiment? Perhaps it is better to show this result with MSE loss rather than CE to be consistent with the theoretical study.
4. On the effect of regularization, how consistent are the empirical observations on different architectures and datasets? Is the effect of needing to train for more epochs when using a smaller regularization taken into account in the interpretation? Specifically, when using 0 WD, typically the training needs to be much longer. Results shown in Fig 3 will be more convincing if the training length aspect is eliminated.

Miscellaneous queries, comments:

1. Theorem 4.1: the result is up to O(norm squared of deviation in optimal solutions with and without perturbation). However, this applies to the statement approximating the actual deviation in the optimal solutions. Is this reasonable?
2. Just before the statement of Cor 2.2, H0 is stated as “already collapsed”. It is better to clarify that this H0 is in fact the ETF/OF solution itself in addition to being collapsed.



**Summary Of The Paper:**

The paper studies neural collapse in DNNs and proposes a modification to the unconstrained features model (UFM) that has been studied by several previous works. In particular, the authors seek to explain a question on the variability of class-wise features (NC1) under the MSE loss. Previous works using the UFM conclude that the features collapse to their class-means, leading to zero variability among the features of examples belonging to a class. Here, the authors, motivated by the fact that in practical neural networks, the last layer features are not truly unconstrained (thus they are a function of the input examples), they propose adding to the UFM a penalty on the features deviating from a fixed features matrix. For this model, the authors theoretically prove that an NC1 metric decreases monotonically along the gradient flow path. Further, with the so-called perturbation analysis, approximate values of the deviation in the optimal feature are given when the fixed feature deviates by a small amount. This result leads to some useful insights on the sensitivity of the optimal features under near-collapse to regularization hyperparameters. Some empirical evidence is provided that correlates with certain insights gained from the theoretical study.

**Summary Of The Review:**

Overall I think the topic is relevant, the author's idea of modeling non-perfect NC is interesting and the paper is contributing something new to the recent literature on neural collapse. I have some question about the actual implications of the result, the practical relevance of the model and the experiments as discussed above. My first impression is positive and looking forward to the discussion.

---

> ### Author Response · Authors · 2022-11-15
> **Authors Response (1/2)**
>
> We thank the reviewer for handling our paper. Below is our point-to-point response to the comments.
>
> 1. *On the distance from zero of NC metrics*
>
> The experiments with practical DNNs and datasets in this paper and also in other papers (e.g., Figure 6 in (Papyan et al., 2020 [1])) show that the NC1 metric reaches a plateau above zero and having more epochs (even with reduced learning rate) do not yield further decrease. The gap from zero seems to depend on the architecture, hyperparameters, dataset complexity, etc.
> Moreover, oftentimes, it seems that other NC metrics, e.g., the metric for NC2, have even more significant gap above zero.
>
> Proving that the global minimizer is not an ``NC point" (i.e., does not have an exact NC structure) requires stop treating the features as free optimization variables without any constraints. For example, in our model, where the features $H$ are penalized by their distance from $H_0$, unless $H_0$ is zero or already a solution of Eq. (1) then the minimizer is not an NC point.
> Practical networks do not optimize the features directly but rather learn a feature mapping $h(\cdot)$. In this case, a simple example where exact NC cannot happen is when two identical samples $x_i$ and $x_j$ have different labels.
>
> *Importantly*, in our paper we do not argue that exact collapse cannot be achieve by any DNN in any setting.
> What do claim is the empirical fact: in typical experiments there exist gaps from exact NC structure.
> Ignoring this empirical fact motivates the study of ``ideal" models, such as all those in the exiting NC theory literature, for which all the minimizers exhibit exact NC structure.
> The main consequence of this trend is that focusing on such models masks the effects of the architecture, hyperparameters, and dataset complexity on the closeness to collapse and limits our understanding. In our model, on the other hand, we do capture the effects of regularization hyperparameters and the depthwise progress of collapse.
>
>
> 2. *Experiments and advantages over the plain UFM*
>
> Our findings on the proposed model (e.g., layer-wise within-class variability decrease and effects of hyperparamters on the minimizer) are theoretical facts that cannot be obtained for the previously studied UFMs (as stated in the paper, the minimizers of these models have exact NC structure regardless the (nonzero) regularization level and they do not capture depthwise behavior).
> In that sense, the weakness of simple UFM compared to our model (which subsumes UFM as a special case) is clear.
>
> Therefore, we assume that the reviewer means more experiments with practical DNNs.
> Regarding the examination of the regularization hyperparameters (for examining correlation with the results of Section 4),
> note that we have 12 settings (4 settings in each of the Figures 3, 4, and 5) with 3 regularization configurations in each setting, and 4 regularization configurations in Figure 6.
> We also present statistical results in Table 1.
> Note also that we use a practical DNN (ResNet18) with practical end-to-end training on practical datasets (CIFAR-10 and MNIST).
> In the final version we will add experiments with another DNN and dataset.
>
> Regarding the examination of layer-wise decrease in NC1 metric (for examining correlation with the results of Section 3), for end-to-end training we have empirical evidence in Figures 4 and 5 (and we refer to demonstration of this behavior in other papers), and for layer-wise training we have Figure 2 where we used a 10-layer MLP (which has a fixed features dimension as required in the layer-wise training and not common in most DNNs).
> We did not provide extensive experimentation for the layer-wise style of training because it is not common in practice; indeed, it seems to lead to worse performance and make the loss minimization harder (as we have observed ourselves and further discuss below).

---

> > ### Author Response · Authors · 2022-11-15
> > **Authors Response (2/2)**
> >
> > 3. *The layer-wise training experiment*
> >
> > We thank the reviewer for this comment. We should have included in Appendix E.1 the following details that have now been added to the revision.
> >
> > We have found that layer-wise training of DNNs (on a practical dataset, e.g., CIFAR-10 that we use here) is significantly harder than end-to-end training in terms of reaching a small loss value. (Presumably, this is the reason that DNNs are typically trained in an end-to-end fashion).
> > Careful configuration of the training procedure was required for reaching considerable low loss (though, still not zero training error) and low NC1 metrics as presented in Figure 2.
> > From our efforts in layer-wise training the 10-layer MLP we observed the following: Adam optimizer worked better than SGD (which is harder to tune); Layer-wise minimization with CE loss has led to lower NC1 metrics; Using no more than 3 epochs per ``outer iteration" allowed reaching lower values for the loss and the NC1 metrics at the deeper layers.
> > Regarding the latter (i.e., more epochs per outer iteration lead to worse optimization results), when there are only one or two hidden layers then the decrease in the loss and the decrease in the NC1 metrics are larger when more epochs are being used. However, when we add in that case more hidden layers,
> > the optimization appears to get stuck at some local minima with higher loss and NC1 metrics compared to what we get with only 3 epochs per outer iteration.
> > As far as we understand, this behavior follows from the (extreme) nonconvexity of the problem.
> >
> > In the final version we will add a layer-wise experiment with MSE loss as well. However, note that even though we prove results for MSE loss (to allow rigorous mathematical analysis), it is in fact a benefit that we present experiments for CE as well, to illustrate that we are not revealing peculiar features of MSE that don't appear in other settings.
> >
> >
> > 4. *Experiments on the effects of regularization*
> >
> > Except for the experiment that has led to Figure 2, all the experiments use typical end-to-end training of DNNs.
> > In all of the figures that are associated with end-to-end training, and Figure 3 is among them, the x-axis is the number of epochs. Thus, it can be seen that the NC metrics reach plateaus in these experiments, so it is not a matter of having more epochs.
> >
> > Regarding the architectures and datasets, we use a practical DNN (ResNet18) and practical datasets (CIFAR-10 and MNIST).
> > In the final version we will add experiments with another DNN and dataset.
> >
> >
> > - *Miscellaneous comment 1*
> >
> > The second-order approximation accuracy follows from using first-order Taylor approximation in the proof.
> > We believe that these requirements of the analysis technique are reasonable since the insights that are gained from this analysis are aligned with practical DNN settings (beyond the scope of the analysis).
> >
> > - *Miscellaneous comment 2*
> >
> > This clarification is added to the revision.
> >
> > - *References*
> >
> > [1] Vardan Papyan, XY Han, and David L Donoho. Prevalence of neural collapse during the terminal phase of deep learning training. Proceedings of the National Academy of Sciences, 117(40): 24652–24663, 2020.

---

### Official Review · Reviewer_uexe · 2022-10-30

**Confidence:** 4
**Correctness:** 4
**Technical Novelty And Significance:** 2
**Empirical Novelty And Significance:** 2
**Recommendation:** 5

**Clarity, Quality, Novelty And Reproducibility:**

Overall, the paper is well-organized and easy to follow. But the notations in section 4 are a little bit confusing. For example, $H_0$, $H^*$, and $\hat H^*$ all refer to the same one. The perturbation result is new and provides some new insights about neural collapse.

**Strength And Weaknesses:**

## Strength:
- This paper provides a new model based on the existing unconstrained features model for understanding neural collapse.
- The new model could potentially provide a further justification for neural collapse observed in practical networks.
- The authors provide analysis based on gradient flow and perturbation analysis for the global solutions of the new model. The analysis shows a reduction in the within-class variability of the output features compared to the predefined input features.
## Weakness:
- The major concern is the rationality behind the proposed new model. In particular, the new model, as shown in eq. (2), is the same as the existing unconstrained features model but with an additional regularizer on the distance to predefined features $H_0$. Why this model can approximate what happened in practice? What do these predefined features $H_0$ represent? If these $H_0$ represent the features before the last-layer features $H$, then in practice, $H$ is obtained as a linear transformation (and plus nonlinear functions) of $H_0$, which may be far away as simply adding regularizer on the distance between $H$ and $H_0$. In any case, more discussions should be provided to support this new model.
- The analysis in Section 3 uses a new measure for neural collapse, $\tilde NC_1$, and shows the decrease of $\tilde NC_1$. But the experimental results are plotted in terms of $NC_1$. Does a decrease of $\tilde NC_1$ imply a decrease of $NC_1$?
- The results in Section 3 only show a decrease in neural collapse, but do not provide a specific amount of decrease.
- The results in Section 4 assume $\beta$ is much larger than 1, which implies that the learned features will be very close to $H_0$. The main result in Theorem 4.1 also suggests this, as eq. (5) implies that the collapsing gap is approximately preserved, i.e.,  $\delta H \approx \delta H_0$. This is not close to what we observed in practice as in Figure 2, where the features become collapse very quickly across layers.

**Summary Of The Paper:**

This paper focuses on the understanding of neural collapse (NC) phenomena of the learned last-layer features and classifiers observed in deep learning classifiers. Most of the existing work provides analysis under the so-called unconstrained features model where the features are viewed as free optimization variables. Motivated by the fact that dee layers cannot arbitrarily modify intermediate features that are far from being collapsed, this paper proposes a slightly different model by forcing the features to stay in the vicinity of predefined features. This is achieved by adding a regularizer to the existing unconstrained features model. The authors provide a perturbation analysis for the new problem with an additional penalty on the distance to predefined features and show that within-class variability of the output features is reduced compared to the predefined input features.

**Summary Of The Review:**

This paper studies a clear and important question about approximate neural collapse observed with practical networks. However, there is a lack of sufficient description of why the proposed approach can be used to explain the approximate neural collapse. For example, it is not clear to me what the term $||H - H_0||_F^2$ represents in a practical network.  It also appears that the proposed approach can not capture the observed phenomena in practice.

---

> ### Author Response · Authors · 2022-11-15
> **(1/2)**
>
> We thank the reviewer for handling our paper. Below is our point-to-point response to the comments.
>
>
> - *The rationality behind the proposed new model*
>
> Thank you for this important comment.
> Indeed, in the model that we consider (Eq. (2)) the relation between $H$, which we interpret as the deepest features, and $H_0$, which we interpret as the preceding (penultimate) features, differs from their explicit relation in many practical DNNs.
> Yet, there exist also networks where it is very reasonable to assume that the deepest features and the penultimate features are close to each other. Specifically, consider the ResNet architecture of (He et al., 2016 [1]) where (under our interpretation of $H$ and $H_0$) the deepest features obey $H = H_0 + r(H_{0})$, where $r(\cdot)$ denotes a residual block. The residual term can potentially be very small if $H_0$ already separates the classes (e.g., it has a ``near NC" structure).
> In fact, in the popular neural ODE framework (Chen et al., 2018 [2]), which is understood as the infinite depth limit of these ResNets, we inherently have that $H \approx H_{0}$.
> Another example where the concept $H \approx H_{0}$ inherently holds is deep equilibrium models (DEQ) (Bai et al., 2019 [3]).
> These practical models provide the rationality for analyzing our model.
> Furthermore, our theoretical results, such as depthwise decrease in the within-class variability, are aligned also with the empirical behavior of DNN architectures beyond the aforementioned examples (e.g., the plain MLP in Figure 2).
> This discussion, which motivates our model, is added to Section 2 and Appendix F of the revision.
>
> To conclude, like all the models in theoretical NC papers, our model in Eq. (2) is much simpler than practical DNNs.
> Nevertheless, it allows us to establish theoretical results (e.g., on the progress of collapse between two levels of features and effects of regularization hyperparameters in the near-NC regime) that cannot be obtained with previous models and empirically correlate with the behavior of practical DNNs.
>
>
> - *The NC1 metrics*
>
> In general, it does not seem that one of the two NC1 metrics ($\tilde{NC}_1(H)$ and ${NC}_1(H)$) is necessarily larger than the other.
> Yet, the important thing (as stated in Section 3) is that the nondegenerate event $\{ \Sigma_W(H) \xrightarrow{} 0, \Sigma_B(H) \nrightarrow{} 0 \}$ occurs iff each of the two metrics converges to zero.
>
> Furthermore, note that *we do plot* $\tilde{NC}_1$ in Figure 2, which demonstrates depthwise decrease of within-class variability in layer-wise training and corroborate the theory of Section 3.
> As we stated in the beginning of the experiments section: We use the $NC_1$ metric in most of our experiments due to its popularity (even though it is less amenable for theoretical analysis).
> If this is critical to the reviewer, we can present also $\tilde{NC}_1$ in other figures. Note though that the goal of the other figures is to corroborate the theory of Section 4, which is not tailored to $\tilde{NC}_1$.
>
>
> - *The results in Section 3*
>
> Our paper is the first to theoretically show decrease in a metric for within-class variability between two levels of features.
> Even reaching a strict decrease, as we do, is a challenging mathematical task (as can be seen in the proofs).
> Trying to quantify the amount of decrease is an interesting direction that appears  to be extremely difficult, and we believe that it is fair to leave it for future research.
>
>
> - *The results and Figure 2*
>
> The large $\beta$ assumption is required for the theoretical analysis.
> Even in our model in Eq. (2) with small/medium $\beta$, the distance metrics of the minimizer $H$ from exact NC is expected to become smaller (e.g., think about coinciding with the plain UFM of Eq. (1) when $\beta\xrightarrow{}0$).
> It is very common that theoretical analysis focuses on a regime that can be analyzed mathematically (e.g., as explained in Section 4, without the large $\beta$ assumption we cannot replace a complicated matrix inversion with a manageable approximation).
>
> Regarding Figure 2, which presents layer-wise training of a practical DNN,
> as we state when we present Figure 2: the decrease in NC1 metrics that is observed in early layers is in fact aligned with the theory in Section 3 that holds for *arbitrary $H_0$*.
> On the other hand,
> recall that the results of Section 4 consider the ``near-NC regime" (where $H_0$ is near a point with NC structure).
> Therefore, the fast decrease in NC1 metrics in early layers in Figure 2 do not show any deviation from our theory in Section 4 since the features in shallow layers are far from being close to exact NC structure.

---

> > ### Author Response · Authors · 2022-11-15
> > **(2/2)**
> >
> > - *Response to the summary*
> >
> > The motivation for the model is further explained in our answer to the reviewer's first comment.
> > The agreement of theory and practical NC behavior is explained in the rest of our answers. As always, the models that are rigorously analyzed in deep learning theory are simplified and cannot capture the DNNs behavior is any possible regime. But our experiments show that the model that we study in this paper (with the term $\|H-H_0\|_F^2$) *does* suffice for reaching new theoretical results and insights that shed light on phenomena of practical DNNs, such as within-class variability decrease as we progress in the level of the features and roles and effects of the regularization hyperparameters in the near collapse regime.
> > Importantly, previous models in the literature fail to explain/capture these phenomena that we can explain with our model.
> >
> >
> > - *References*
> >
> > [1] He, Kaiming, Xiangyu Zhang, Shaoqing Ren, and Jian Sun. "Identity mappings in deep residual networks." In European conference on computer vision, pp. 630-645. Springer, Cham, 2016.
> >
> > [2] Chen, Ricky TQ, Yulia Rubanova, Jesse Bettencourt, and David K. Duvenaud. "Neural ordinary differential equations." Advances in neural information processing systems 31 (2018).
> >
> > [3] Bai, Shaojie, J. Zico Kolter, and Vladlen Koltun. "Deep equilibrium models." Advances in Neural Information Processing Systems 32 (2

---

### Decision · Program_Chairs · 2023-01-20

**Decision:**

Reject

**Justification For Why Not Higher Score:**

The authors provided responses to the initial reviews, which however did not sufficiently resolve some of the reviewers' concerns.

**Justification For Why Not Lower Score:**

NA

**Metareview: Summary, Strengths And Weaknesses:**

This work investigates the neural collapse phenomenon under a modified version of a previously studied unconstrained feature model in which the features remain close to predefined values.

* Strengths are the progression over simplistic models and contributing to the theoretical description of neural collapse.
* Weaknesses are insufficient justification for the proposed model and assumptions, limited explanatory power of the results, and concerns on limited innovation.

After the discussion period some concerns could be clarified, but other concerns remained, particularly about certain parts of the analysis, the justification of the proposed model, and the significance of the contribution. I find that the article contains promising ideas but also share some of the reviewers' concerns and find it would have been stronger by more clearly motivating and further developing the results in relation to practical settings.



**Summary Of Ac-Reviewer Meeting:**

NA